# GABAergic neurons in the rostromedial tegmental nucleus are essential for rapid eye movement sleep suppression

Ya-Nan Zhao [1,2], Jian-Bo Jiang [1,2], Shi-Yuan Tao[1], Yang Zhang[1], Ze-Ka Chen[1], Wei-Min Qu [1] ✉, Zhi-Li Huang [1] ✉ & Su-Rong Yang [1] ✉

Rapid eye movement (REM) sleep disturbances are prevalent in various psychiatric disorders. However, the neural circuits that regulate REM sleep remain poorly understood. Here, we found that in male mice, optogenetic activation of rostromedial tegmental nucleus (RMTg) GABAergic neurons immediately converted REM sleep to arousal and then initiated non-REM (NREM) sleep. Conversely, laser-mediated inactivation completely converted NREM to REM sleep and prolonged REM sleep duration. The activity of RMTg GABAergic neurons increased to a high discharge level at the termination of REM sleep. RMTg GABAergic neurons directly converted REM sleep to wakefulness and NREM sleep via inhibitory projections to the laterodorsal tegmentum (LDT) and lateral hypothalamus (LH), respectively. Furthermore, LDT glutamatergic neurons were responsible for the REM sleep-wake transitions following photostimulation of the RMTg[GABA]-LDT circuit. Thus, RMTg GABAergic neurons are essential for suppressing the induction and maintenance of REM sleep.

Based on changes in electroencephalograms (EEG), electromyograms (EMG), and sympathetic activity, mammalian sleep consists of two distinct states. Compared with the large amplitude of slow-wave activity in non-rapid eye movement (NREM) sleep, rapid eye movement (REM) sleep is characterized by low-voltage, fast electroencephalographic activity, loss of muscle tone, intermittent muscle twitches, and concomitant autonomic and respiratory activation[1–3]. Although the normal function of the body is generally a result of the complementary effects of these two types of sleep, a large amount of evidence shows that REM sleep plays a pivotal role in processing emotional memories and regulating emotional responses[4–6]. Abnormal REM sleep is often observed in patients with psychiatric disorders. For example, REM sleep disturbances are recognized as etiological mechanisms in the occurrence, progression, and prognosis of post-traumatic stress disorder (PTSD) and depression. Shortened REM sleep latency and increased REM sleep duration are highly prevalent in such patients[7–11]. However, the neural systems that regulate REM sleep are not well understood, thereby restricting the renewal of theories and

treatments for REM sleep disorders. Hence, the identification of neurons and circuits that regulate REM sleep is of great scientific interest and clinical importance.

The dorsal pons has long been known to play a key role in the generation of REM sleep. In addition, several studies have revealed that many brain regions, such as the hypothalamus, midbrain, and medulla, are also involved in REM sleep control[12–14]. The rostromedial tegmental nucleus (RMTg), predominantly composed of GABAergic neurons, receives projections from widespread brain structures, including the lateral habenula (LHb). After integrating the convergent signals, the RMTg participates in motor control and emotion processes such as punishment learning and aversive valence encoding. To date, the RMTg has been recognized as a major "brake" on motivated behavior[15–17]. Aside from heavy output to the midbrain dopaminergic neurons, neuroanatomical studies have also shown that multiple nuclei, including the laterodorsal tegmentum (LDT), pedunculopontine tegmentum (PPT), lateral hypothalamus (LH), dorsal raphe (DR), and ventrolateral periaqueductal gray (vlPAG), are innervated by

[1]Department of Pharmacology, School of Basic Medical Sciences; State Key Laboratory of Medical Neurobiology and MOE Frontiers Center for Brain Science; Institutes of Brain Science, Fudan University, Shanghai 200032, China. [2]These authors contributed equally: Ya-Nan Zhao, Jian-Bo Jiang. ✉e-mail: quweimin@fudan.edu.cn; huangzl@fudan.edu.cn; sryang@shmu.edu.cn

RMTg GABAergic neurons[18–20], and these are believed to be important for REM sleep regulation[3]. In addition, in our previous study, we found that non-specific activation of neurons in the rat RMTg dramatically increased NREM sleep; however, it also dramatically decreased REM sleep[21]. Based on this, we hypothesized that the RMTg is responsible for REM sleep suppression.

In this study, optogenetic and chemogenetic approaches combined with EEG recordings were used to investigate the sufficiency and necessity of GABAergic neurons, the primary neuronal type in the RMTg, for suppressing REM sleep. We then performed optrode recordings to characterize the spiking activity of these neurons across sleep–wake states. We further identified the descending projections and neural circuits of these cell populations in control of REM sleep using anterograde tracing techniques, patch–clamp recordings, RNA interference, optogenetics, and EEG recordings. Thus, the present study revealed RMTg GABAergic neurons as a cell population involved in the suppression of REM sleep and explored the potential circuit mechanisms.

## Results

### Chemogenetic activation of RMTg GABAergic neurons suppresses REM sleep in VGAT-Cre mice

To explore the roles of RMTg cell-specific neurons in REM sleep regulation, we used transgenic mice combined with chemogenetics and optogenetics. However, no clear location of the RMTg in the mouse atlas was identified. It has been reported that the transcription factor Foxp1 is highly expressed in the RMTg rather than in the adjacent area, including the ventral tegmental area (VTA), as demonstrated by both RNA sequencing analysis and Foxp1-immunohistochemical experiments[22,23]. Accordingly, we labeled Foxp1 immunoreactive neurons and found that Foxp1 was expressed at higher levels in the RMTg than in the surrounding regions (Fig. S1a, left). The relative location of the RMTg was placed on the mouse atlas according to the range of Foxp1 expression in the mesopontine, which spreads from bregma at −3.40 to −4.84 mm (Fig. S1b, c). Moreover, fluorescent double-label staining showed that in the mouse RMTg, 96% of Foxp1-expressing neurons were colocalized with GABA, and 93% of GABA-positive neurons co-expressed Foxp1 (Fig. S1a, right). As a result, Foxp1 was used as a specific molecular marker to define the location of the RMTg, and GABAergic neurons in the RMTg were manipulated using vesicular GABA transporter (VGAT)-Cre mice in this study.

First, we tested the function of RMTg GABAergic neurons in REM sleep regulation. Adeno-associated virus (AAV) vectors containing excitatory-modified muscarinic Gq protein-coupled receptors (hM3Dq) were bilaterally microinjected into the RMTg of VGAT-Cre mice (Fig. 1a, Fig. S2). Confocal images of double labeling (yellow) with mCherry (red) and GABA (green) immunofluorescence showed that 92% of hM3Dq/mCherry-positive neurons co-expressed GABA (Fig. 1b), validating the specificity of this targeting strategy for RMTg GABAergic neurons. Using whole-cell current clamp recordings, the application of clozapine-N-oxide (CNO, 5 μM), a specific ligand for hM3Dq receptors, increased the firing rates of the action potentials of the hM3Dq/mCherry-positive neurons and depolarized them (Fig. 1c, d). This confirmed that the chemogenetic system used in this study excited RMTg GABAergic neurons in VGAT-Cre mice.

When RMTg GABAergic neurons were specifically activated by CNO injection at a dose of 1 mg/kg, the mice showed a significant decrease in REM sleep and an increase in NREM sleep compared with the saline group, which was consistent with the effects when the rat RMTg neurons were non-specifically activated[21]. During the 8-h post-CNO injection period (9:00–17:00), the animals' REM sleep amount significantly decreased from 38.7 ± 4.4 to 19.2 ± 4.7 min (Fig. 1e, f). The sleep architecture analysis revealed no significant decrease in the mean duration of REM sleep episodes in the mice administered CNO. However, the number of total episodes of REM sleep after activation of

the RMTg GABAergic neurons decreased, which was evidenced by fewer REM sleep bouts with durations between 64 and 128 s. The diminished episodes of REM sleep resulted from fewer conversions from NREM to REM sleep, which resulted in fewer conversions from REM sleep to wakefulness (Fig. 1g–j). These results indicated that activation of RMTg GABAergic neurons remarkably inhibited the generation of REM sleep bouts, which was associated with a lower amount of REM sleep. However, when RMTg GABAergic neurons were activated during the active period, REM sleep did not significantly decrease, which may be that there is less REM sleep in the dark phase (Fig. S3).

### Optogenetic activation of RMTg GABAergic neurons terminates REM sleep and suppresses REM sleep initiation in VGAT-Cre mice

After the long-term effects on REM sleep regulation were observed by chemogenetic manipulation of RMTg GABAergic neurons[24], we applied optogenetics with high temporal resolution at the millisecond level to investigate the immediate effects[25]. AAVs encoding channelrhodopsin 2 (ChR2) fused to an enhanced red fluorescent protein (mCherry) or AAVs encoding only mCherry were delivered to the bilateral RMTg in VGAT-Cre mice (Fig. 2a, b, Fig. S2). In vitro experiments showed that blue light stimulation evoked ChR2-expressing neurons to produce action potentials. Sustained and stable discharge of RMTg GABAergic neurons was induced by a laser with a frequency ranging from 1 to 40 Hz (Fig. 2c). These results confirmed that ChR2 was driven by blue light with high temporal precision.

We next stimulated RMTg GABAergic neurons in ChR2-expressing mice with 10 ms pulses of blue light at 30 Hz for 120 s during REM sleep, in which the stimulation always began at least 16 s after the onset of REM sleep. Laser stimulation of GABAergic neurons in the bilateral RMTg immediately produced transitions from REM sleep to wakefulness and then NREM sleep in ChR2 mice, whereas the control mice continually maintained REM sleep (Fig. 2d, e). Statistical analysis of all transitions of REM sleep to other brain states showed that the latency of REM sleep transitions decreased from 77.1 ± 4.4 s in the mCherry group to 7.8 ± 1.5 s in the ChR2 group following blue light stimulation (Fig. 2f). During the 120-s photostimulation period, the cumulative probabilities of REM sleep termination, NREM sleep, and wake initiation were higher in ChR2 mice than in mCherry mice (Fig. 2g). As the frequency of blue light illumination increased from 5 Hz, 10 Hz, 20 Hz, 30 Hz, and 40 Hz, the REM sleep conversion latency decreased by 3.4%, 22.2%, 74.6%, 90.3%, and 91.5%, respectively, in the ChR2 group compared to the mCherry group. When RMTg GABAergic neurons were stimulated by blue light at 30 Hz and 40 Hz, the latency of REM sleep conversion was less than 10 s (Fig. S4). Furthermore, the in vivo study showed that the spike frequency during REM sleep raised up to 30 Hz. Therefore, 30 Hz was used in the optogenetic experiments. The photostimulation duration of 120 s that we used was based on the finding that there were significant differences in REM sleep transition latency between the ChR2 and mCherry groups, regardless of the stimulation duration (30, 60, 120, or 300 s). However, 120-s stimulations could induce a relatively longer NREM sleep duration from the awake state than light stimulation with other durations (Fig. S5).

Additionally, we observed that blue light stimulation of RMTg GABAergic neurons during NREM sleep, which continued for at least 60 s, consolidated the NREM sleep state and completely blocked REM sleep generation (Fig. 2h, i). Conversely, during wakefulness, laser stimulation converted arousal into NREM sleep (Fig. S6). After the 120-s photostimulation of the RMTg GABAergic neurons expressing ChR2, there was a rebound following REM sleep suppression but not following wakefulness suppression (Fig. S7). These results showed that optogenetic activation of RMTg GABAergic neurons inhibited the initiation and maintenance of REM sleep.

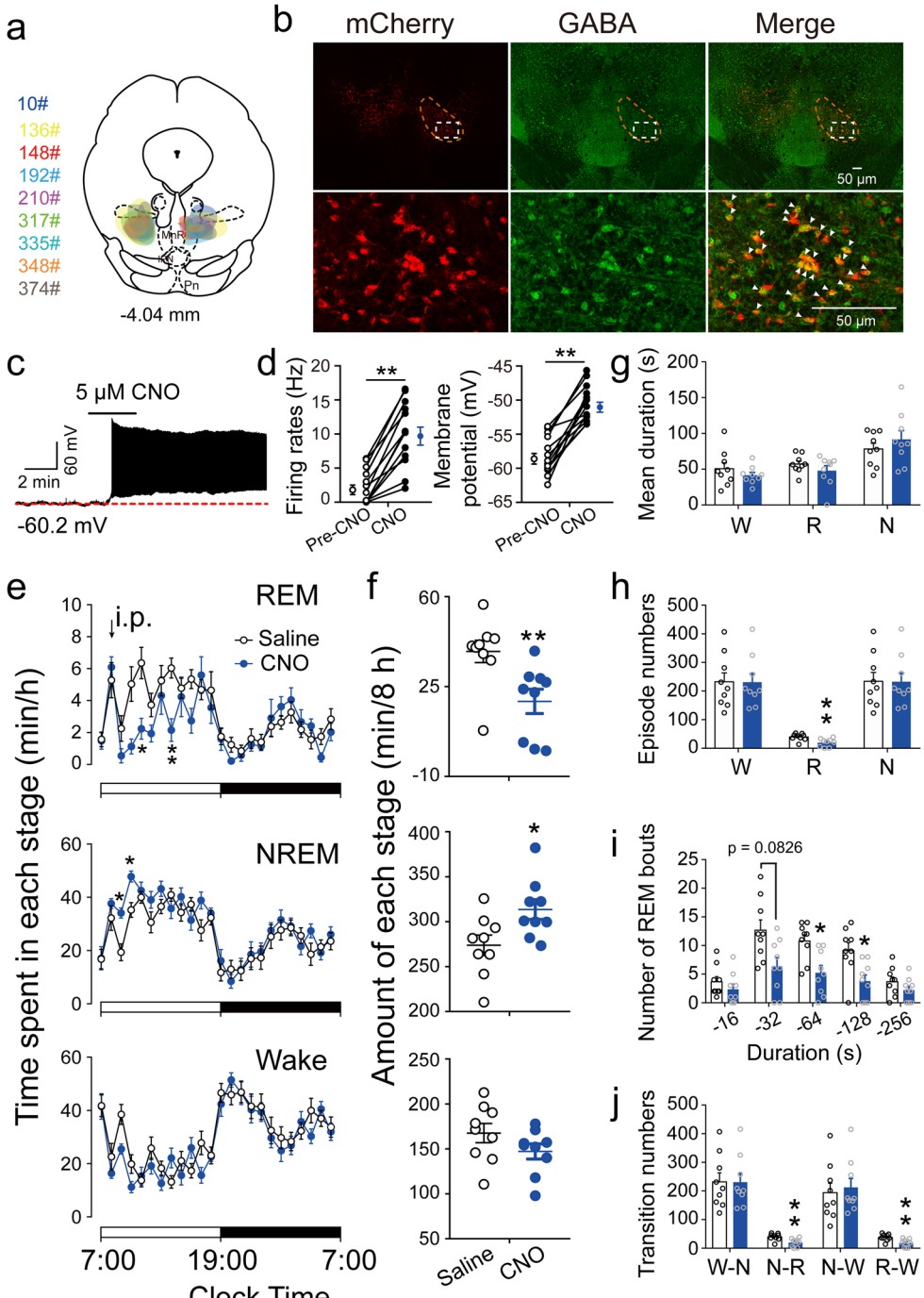

**Fig. 1 | Chemogenetic activation of RMTg GABAergic neurons suppressed REM sleep. a** A coronal section shows the superimposed injection sites of adeno-associated virus (AAV) encoding hM3Dq fused to an enhanced red fluorescent protein (mCherry) reporter into the bilateral RMTg in nine VGAT-Cre mice. **b** Representative photomicrographs of RMTg GABAergic neurons from a mouse microinjected with AAV vectors encoding hM3Dq. The mCherry (red) and GABA-immunolabeling (green) indicate hM3Dq- and GABA-expressing neurons, respectively, and the yellow image depicts merged neurons. Top: lower magnification of the images. The dashed circle area shows the location of the RMTg. Bottom: higher magnification of the dashed square region in the top corresponding images. **c** A typical trace recorded from an hM3Dq$^+$ neuron in the RMTg during appication of Clozapine-N-oxide (CNO). **d** CNO significantly increased the firing rates ($T_{12} = 8.446$, $p < 0.0001$) and membrane potential ($T_{12} = 9.217$, $p < 0.0001$) of RMTg hM3Dq-expressing neurons ($N = 13$ cells from 5 mice). **e, f** After administration of

saline or CNO at 9:00, the hourly average amount of each stage (REM sleep: $F_{1,16} = 9.158$, $p = 0.0080$; NREM sleep: $F_{1,16} = 6.395$, $p = 0.0223$; wake: $F_{1,16} = 2.254$, $p = 0.1527$) (**e**) and the total amount of each stage (REM sleep: $T_8 = 3.662$, $p = 0.0064$; NREM sleep: $T_8 = 2.410$, $p = 0.0425$; wake: $T_8 = 1.662$, $p = 0.1351$] during the 8-h period (9:00–17:00) (**f**). **g–j** Sleep–wake architecture during the 8-h post-injection period (9:00–17:00), including mean duration (**g**), episode number (REM sleep: $T_8 = 5.438$, $p = 0.0006$) (**h**), number of REM sleep bouts with different durations ($F_{1,16} = 10.34$, $p = 0.0054$) (**i**), and conversions between W (wakefulness), N (NREM sleep), and R (REM sleep) (N–R: $T_8 = 5.383$, $p = 0.0007$; R–W: $T_8 = 5.041$, $p = 0.001$) (**j**). *$p < 0.05$, **$p < 0.01$. Statistics by two-way repeated ANOVA followed by a post hoc Sidak test (**e, i**) and by paired $t$ test (**f–h, j**). Data represent mean ± SEM. IPN interpeduncular nucleus, MnR median raphe nucleus, Pn pontine reticular nucleus. Source data are provided as a Source Data file.

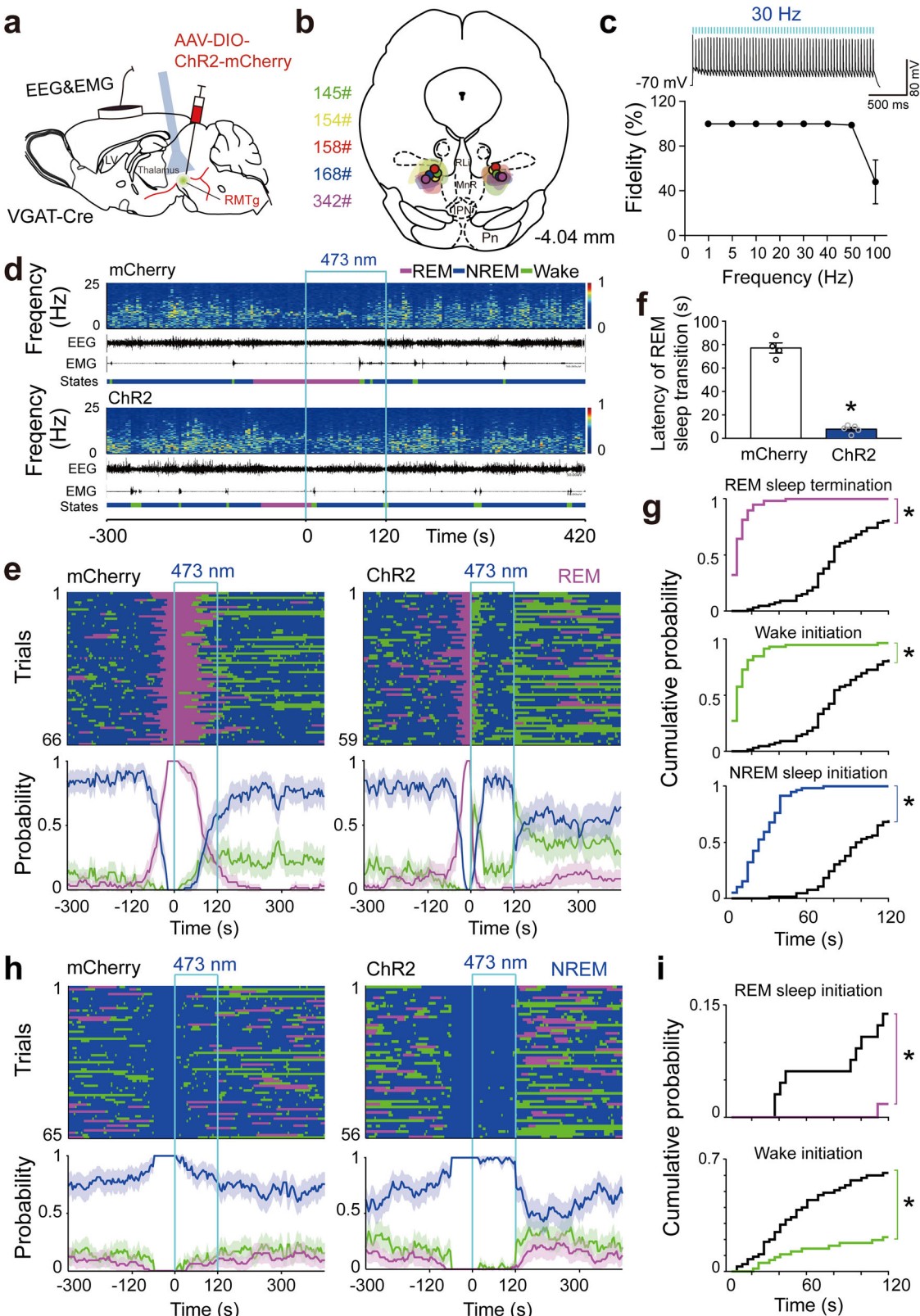

## Optogenetic inactivation of RMTg GABAergic neurons completely converts NREM into REM sleep and inhibits REM sleep transitions in VGAT-Cre mice

To investigate the physiological roles of RMTg GABAergic neurons in REM sleep regulation, we used chemogenetics to inactivate these neurons and observed their effects on sleep-wake behaviors. When

GABAergic RMTg neurons were inactivated by CNO administration (Fig. S2, S8a–c), the mice showed a decrease in REM and NREM sleep and a corresponding increase in awake duration during the 2-h post-CNO injection period (9:00–11:00). However, within the 4h period (13:00–17:00), the amount of REM sleep significantly increased from $19.4 \pm 2.3$ to $25.4 \pm 1.8$ min but there was no increase in

**Fig. 2 | Optogenetic activation of RMTg GABAergic neurons terminated REM sleep and inhibited REM sleep induction. a** Schematic illustration of virus injection. Adeno-associated virus (AAV) encoding channelrhodopsin 2 (ChR2) fused to an enhanced red fluorescent protein (mCherry) reporter was introduced into the bilateral RMTg in VGAT-Cre mice. **b** The coronal section shows the superimposed virus-injected area in five mice. **c** A ChR2-mCherry expressing RMTg GABAergic neuron showed spiking following 30 Hz blue light illumination in current–clamp mode (top). Fidelity analysis of the action potential evoked by stimulus at frequencies ranging from 1 Hz to 100 Hz (bottom). **d** Representative EEG and EMG traces and corresponding heat map of EEG power spectrum in different sleep–wake states from a mouse in the mCherry (top) and ChR2 (bottom) group, respectively. Blue line, laser stimulation (30 Hz, 120 s, 10 ms per pulse). **e, h** Sleep–wake state changes (top) and probability of brain state transitions (bottom) after

photostimulation was applied during REM sleep lasting no less than 16 s before laser on (**e**) and NREM sleep lasting no less than 60 s before laser on (**h**) in all trials from mice in the mCherry (left) and ChR2 (right) groups, respectively. Shading indicates 95% confidence intervals. **f** Mean latency of REM sleep transitions to wakefulness or NREM sleep after blue laser stimulation during REM sleep in the first eight trials per mouse. $p = 0.0159$, Mann–Whitney $U$ test. **g, i** During 120 s laser stimulation, the cumulative probability for REM sleep termination, wake, and NREM sleep initiation when laser illuminated during REM sleep (**g**) and REM sleep and wake initiation when laser illuminated during NREM sleep (**i**). Colorful stairs, ChR2 group. Black stairs, mCherry group. Data represent mean ± SEM. *$p < 0.05$ vs. mCherry group, Kolmogorov–Smirnov test. ChR2: $n = 5$, mCherry: $n = 4$. Source data are provided as a Source Data file.

wakefulness (Fig. S8d, e). The promotion of REM sleep was caused by a longer mean duration of REM sleep, because there was no significant increase in REM sleep episodes and conversions among brain states (Fig. S8f–h). These results suggested that RMTg GABAergic neurons may inhibit REM sleep maintenance.

To observe whether the sleep drive caused by sleep deprivation (SD) induces REM sleep rebound, the VGAT-Cre mice with AAVs encoding hM4Di fused to mCherry in the bilateral RMTg were deprived of sleep by gentle handling for 2 h from 9:00 to 11:00. We observed that the 2-h SD after the saline injection significantly decreased the amount of REM and NREM sleep and increased wakefulness, which was similar to the results obtained in mice expressing hM4Di in RMTg GABAergic neurons after CNO administration without SD. However, within 2 h after the SD (11:00–13:00), the amount of REM sleep was still significantly lower than that of the control group. During the following 4-h period (13:00–17:00), the SD mice showed no rebound of REM sleep (Fig. S9). Therefore, when RMTg GABAergic neurons were inhibited by CNO injection in mice, the increase in REM sleep after CNO injection was not due to REM sleep deficiency.

Given that the optogenetic inhibitory method provides an opportunity to explore the necessity of targeted neurons in behavior control, we selectively inactivated RMTg GABAergic neurons using AAVs encoding archaerhodopsin-3 (Arch), which can mediate powerful silencing of neural activities through outward proton pumps following yellow light illumination[26]. AAVs encoding Arch fused to an enhanced green fluorescent protein (eGFP) or AAVs encoding only eGFP were microinjected into the bilateral RMTg of VGAT-Cre mice (Fig. 3a, b, Fig. S2). Whole-cell clamp recordings showed that constant laser irradiation (561 nm) immediately and significantly decreased the membrane potential and produced outward currents in the green Arch-expressing neurons in the RMTg (Fig. 3c, d), indicating the inhibitory effects of yellow illumination of RMTg GABAergic neurons in mice after AAV-Arch-eGFP delivery.

The in vivo studies found that Arch-mediated inactivation of RMTg GABAergic neurons rapidly terminated NREM and initiated REM sleep. In contrast, the control mice with AAV-eGFP microinjections showed no change in NREM sleep during the 60-s laser stimulation (Fig. 3e). In the 51 tests performed in the mice that received AAV-Arch, the conversion rate from NREM sleep to REM sleep with 60-s light stimulation was 100%. However, in the mice that received AAV-eGFP, the transition from NREM to REM sleep occurred in only two of the 60 tests with 60-s light stimulation (Fig. 3f–h). Additionally, silencing RMTg GABAergic neurons during REM sleep prolonged the duration of REM sleep. In the 60 trials performed in Arch-eGFP mice, REM sleep transitions (conversions of REM sleep to NREM sleep or to the awake state) were observed in only two trials with 60-s photostimulation. However, in the 59 trials in the eGFP control animals, REM sleep transitions were observed in 44 trials with 60-s photostimulation. The average latency of REM sleep transitions to other two brain states increased from 50.4 ± 7.3 s in the eGFP group to 114.2 ± 5.5 s in the Arch-eGFP group (Fig. 3i, j). Taken together, these results indicated

that RMTg GABAergic neurons are essential for the inhibition of both REM sleep initiation and maintenance.

## Neuronal activity of RMTg GABAergic neurons as measured by optrode recordings in the sleep–wake states of VGAT-Cre mice

Given that neuronal activity is closely related to brain states, we used the optogenetic tagging and optrode recording to measure the activity changes in RMTg GABAergic neurons on a timescale of milliseconds across spontaneous sleep–wake cycles in freely moving mice[27]. After stereotaxic infusion of Cre-dependent AAVs encoding ChR2-mCherry into the RMTg of VGAT-Cre mice, we implanted a self-designed optetrode to record the spiking activity of RMTg GABAergic neurons and electrodes to simultaneously record EEG/EMG signals. Laser stimulations at 30 Hz with 10 ms per pulse were applied for 10 s. In the RMTg of VGAT-Cre mice, we identified 12 units that showed reliable laser-evoked spiking with latencies less than 5 ms; thus, these were identified as GABAergic neurons (Fig. 4a, b). Unexpectedly, we found that RMTg GABAergic neurons fired more frequently during both REM sleep and wakefulness than during NREM sleep (Fig. 4c). Our analysis of state transitions showed a rapid and strong increase in activity during the conversions from NREM sleep to wakefulness, as well as a decrease during the conversions from wakefulness to NREM sleep and from REM sleep to an awake state (Fig. 4d). Interestingly, the activity of RMTg GABAergic neurons was the lowest at the onset of REM sleep and then gradually increased until it reached a higher frequency near the end of REM sleep. Linear regression revealed that the increase in firing frequency within the first 30 s after REM sleep onset fit a linear equation ($R^2 = 0.819$, $p < 0.01$) (Fig. 4e), which indicated a reciprocal modulation of REM sleep[28,29].

The analysis of firing rates across distinct sleep–wake states revealed two clusters of RMTg GABAergic neurons firing at high and low rates during NREM sleep. Further autocorrelation and coherence analysis illustrated that the firing pattern of RMTg GABAergic neurons during NREM sleep differed from those during wakefulness and REM sleep. The discharge frequency of RMTg GABAergic neurons peaked at 20 and 240 ms during NREM sleep, showing a kind of phasic firing pattern. Moreover, the spiking frequency peaks corresponded to the EEG delta oscillations (Fig. S10).

## RMTg GABAergic neurons promoted REM sleep transitions via LDT and LH circuits

To test the neural circuits of RMTg GABAergic neurons that promote REM sleep conversions, we first observed the targeted projections of these neurons using Cre-inducible AAV-hSyn-DIO-eGFP. The anterograde tracing of RMTg GABAergic axons revealed projections to multiple brain regions such as the VTA, substantia nigra compacta (SNc), LHb, LDT, LH, and DR (Fig. S11). Such broad innervations are in agreement with previous findings in rats[18,19]. As the VTA/SNc, LDT, and LH are densely innervated by RMTg GABAergic axons and previous studies have shown the importance of the LDT and LH in REM sleep regulation[30], and as growing evidence revealed the wake-regulatory

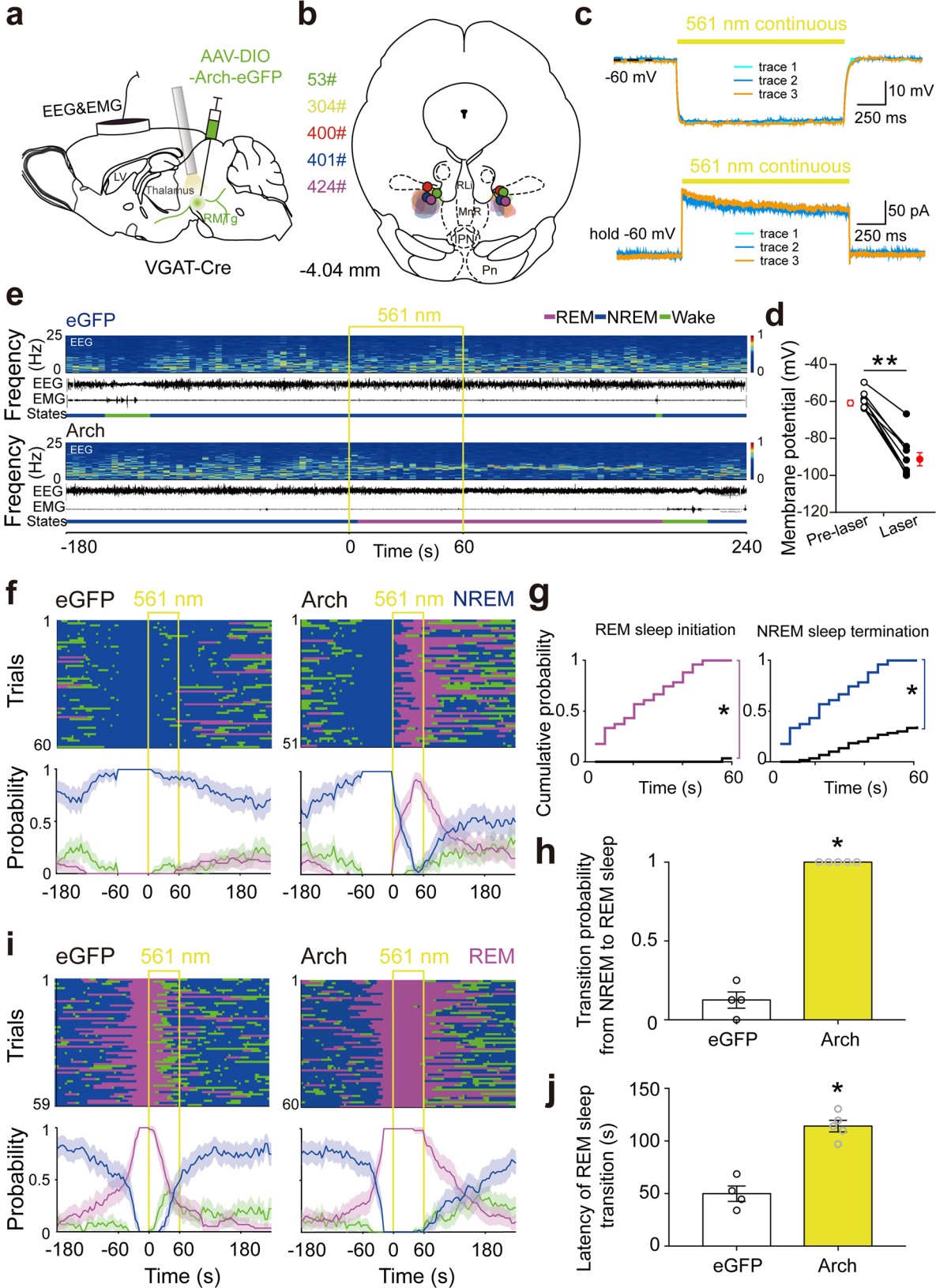

role of dopaminergic neurons[31,32], we investigated whether targeting LDT/LH is implicated in REM sleep inhibition induced by RMTg GABAergic neurons.

ChR2 was expressed in RMTg GABAergic neurons with optical fibers targeting the LDT or LH (Fig. 5a, g, Fig. S2). After introducing

AAV-ChR2-mCherry into VGAT-Cre mice for 3 weeks, blue laser illumination at 473 nm selectively activated ChR2-expressed RMTg GABAergic nerve endings projecting to the LDT or LH, and yellow laser illumination was used as a control. Stimulation of RMTg GABAergic terminals in the LDT with blue light lasers for 120 s at 30 Hz with 10 ms

**Fig. 3 | Optogenetic inactivation of RMTg GABAergic neurons induced REM sleep generation and inhibited REM sleep transition. a** Schematic illustration of virus injection. Adeno-associated virus (AAV) encoding archaerhodopsin-3 (Arch) fused to an enhanced green fluorescent protein (eGFP) reporter was introduced into the bilateral RMTg. **b** The coronal section shows the superimposed virus-injected area in five mice. **c** Yellow laser stimulation induced hyperpolarization in current−clamp mode (top) and outward current in voltage−clamp mode with voltage holding at −60 mV (bottom) in an Arch-eGFP expressing RMTg GABAergic neuron. **d** The average membrane potential ($T_8 = 11.32$, $p < 0.0001$) of RMTg Arch-expressing neurons ($N = 9$ cells from 3 mice) was significantly decreased by yellow laser illumination. The results from each cell are shown on the scatter plot. **e** Representative EEG and EMG traces and the corresponding heat map of the EEG power spectrum in different sleep–wake states from a mouse in the eGFP (top) and Arch (bottom) group, respectively. Yellow line, laser stimulation (constant 60 s).

**f, i** Sleep−wake state changes (top) and probability of brain state transitions (bottom) after photostimulation was applied during NREM sleep lasting no less than 60 s before laser on (**f**) and REM sleep lasting no less than 16 s before laser on (**i**) in all trials from mice in the eGFP (left) and Arch (right) groups, respectively. Shading indicates 95% confidence intervals. **g, h** During 60 s laser stimulation illuminated during NREM sleep, the cumulative probability of REM sleep initiation and NREM sleep termination (**g**) and mean probability of NREM to REM sleep transition in the first eight trials per mouse (**h**). Colorful stairs, Arch group. Black stairs, eGFP group. *$p < 0.05$ vs. eGFP group, Kolmogorov−Smirnov test (**g**). **j** Mean latency of REM sleep transitions to wakefulness/NREM sleep after yellow laser stimulation during REM sleep in the first eight trials per mouse. $p = 0.0159$ vs. eGFP, Mann−Whitney $U$ test (**h, j**). Data represent mean ± SEM. Arch: $n = 5$, eGFP: $n = 4$. Source data are provided as a Source Data file.

pulses resulted in an immediate transition from REM sleep to the awake state; this state quickly transitioned to NREM sleep, but no rapid transition was observed in the control group following yellow light illumination (Fig. 5b, c). Following blue light stimulation, the latency of REM sleep transition to arousal or NREM sleep was 10.2 ± 2.0 s, whereas the latency of REM sleep transitions was 93.4 ± 8.2 s in control mice illuminated with yellow light (Fig. 5d). During the 120-s photo-stimulation period, REM sleep termination and wake initiation were facilitated, followed by an increase in NREM sleep initiation (Fig. 5e, f). Surprisingly, activation of the RMTg$^{GABA}$ to the LH circuit (Fig. 5g) immediately and directly converted REM to NREM sleep (Fig. 5h, i). Following blue and yellow light stimulation of the RMTg GABAergic nerve endings in the LH, the latency of the REM sleep transitions was 6.4 ± 1.1 s and 72.6 ± 2.3 s, respectively (Fig. 5j). Blue light stimulation during the 120-s period produced a significant increase in REM sleep termination and NREM sleep initiation, but no significant increase in wake initiation (Fig. 5k, l). Thus, our optogenetic manipulations revealed two distinct pathways controlling REM sleep transitions: activation of the RMTg$^{GABA}$ to LDT circuit facilitated REM sleep transition to brief wakefulness followed by NREM sleep with a short latency, whereas the RMTg$^{GABA}$ to LH circuit facilitated REM sleep transition to NREM sleep directly.

Notably, during NREM sleep, blue laser stimulation of the RMTg$^{GABA}$-LDT or RMTg$^{GABA}$-LH circuit diminished REM sleep initiation and consolidated NREM sleep (Fig. S12a, c), whereas blue laser stimulation during wakefulness enhanced the transitions to NREM sleep (Fig. S12b, d).

### LDT glutamatergic neurons are necessary for REM sleep−wake transitions caused by activation of the RMTg$^{GABA}$-LDT circuit

It is now known that both the RMTg$^{GABA}$-LDT and RMTg$^{GABA}$-LH circuits are implicated in the suppression of REM sleep generation, as well as the maintenance induced by laser stimulation of RMTg GABAergic neurons. Note that the RMTg$^{GABA}$-LDT circuit is responsible for the REM-wake transitions, which is important for rodents to stay alert while sleeping. The LDT is composed of a heterogeneous population of cholinergic, GABAergic, and glutamatergic neurons, which appear to be the most active during REM sleep and arousal[33,34]. However, the type of cells in the LDT that control REM sleep−wake transitions upon blue laser-mediated activation of RMTg GABAergic terminals is currently unknown.

First, we used AAVs encoding cell-specific promoters or Cre-dependent AAVs encoding fluorescent proteins combined with patch-clamp recordings to test the functional connections from RMTg GABAergic neurons to different types of neurons in the LDT. Cre-inducible AAVs expressing ChR2-mCherry were delivered into the RMTg to express ChR2 on GABAergic nerve endings with red fluorescence, whereas AAVs expressing ChAT-eGFP, DIO-eGFP, or both ChAT-eGFP and DIO-eGFP were introduced into the LDT to mark cholinergic (green), GABAergic (green), and glutamatergic (not green) neurons in

VGAT-Cre mice for patch clamp recording (Fig. 6a, f, k). We found dense mCherry$^+$ RMTg GABAergic terminals in the LDT, which demonstrated anatomical connections between RMTg GABAergic neurons and LDT neurons (Fig. 6b, g, l). In the whole-cell voltage−clamp mode, the blue light pulses of the RMTg GABAergic axon terminals in the LDT evoked inhibitory postsynaptic currents (IPSCs) of the green cholinergic, GABAergic, or colorless glutamatergic neurons. These IPSCs were not abolished by the application of 6-nitro-7-ulphamoylbenzo(f)-quinoxaline-2,3-dione (NBQX, an AMPA receptor antagonist) together with d-(-)−2-amino-5-phosphonopentanoic acid (APV, an NDMA receptor antagonist); however, they were completely abolished by application of picrotoxin (PTX, a GABA$_A$ receptor antagonist) (Fig. 6c, h, m). This indicated that the responses were mediated by GABA release from the RMTg GABAergic axon terminals. Moreover, the light-evoked IPSCs in LDT cholinergic, GABAergic, and glutamatergic neurons were produced in less than 5 ms, indicating a direct inhibitory synaptic connection (Fig. 6d, i, n). The proportion of recorded LDT cholinergic, GABAergic, and glutamatergic neurons that responded to photostimulation of ChR2-expressing RMTg GABAergic terminals was 86%, 75%, and 59%, respectively (Fig. 6e, j, o). After recording, we added biocytin to the pipette solution and performed immunostaining to identify the cell types in the recorded LDT neurons. The indigo cells that merged with eGFP and biocytin were cholinergic or GABAergic neurons, and the violet cells without eGFP expression were glutamatergic neurons (Fig. 6b, g, l).

In this study, we found that inhibitory synaptic connections existed between RMTg GABAergic neurons and cholinergic, GABAergic, and glutamatergic neurons in the LDT. In addition to REM sleep regulation, studies have shown that LDT neurons are active during the awake state. However, optogenetic activation of cholinergic LDT neurons does not induce significant behavioral arousal[35]. As a result, we speculated that RMTg GABAergic neurons may disinhibit LDT glutamatergic neurons to promote REM sleep−wake conversions. To test this hypothesis, a mixture of AAV-DIO-ChR2-mCherry and AAV-ChAT-eGFP was introduced into the LDT of VGAT-Cre mice (Fig. 6p). In the LDT, the neurons expressing mCherry and eGFP were GABAergic and cholinergic neurons, respectively (Fig. 6q). The neurons that expressed neither mCherry nor eGFP were assumed to be glutamatergic neurons and were patch-clamped. We found that 61% of the assumed LDT glutamatergic neurons had direct synaptic inhibitory connections with the GABAergic neurons in the LDT (Fig. 6r−t). Therefore, activation of RMTg GABAergic neurons exerts disinhibitory effects on LDT glutamatergic neurons.

Next, we tested whether activation of LDT glutamatergic neurons is necessary for the transitions from REM sleep to wakefulness. Cre-dependent AAVs encoding ChR2-eGFP and shVglut2-mCherry were introduced into the bilateral RMTg and LDT, respectively, while optical fibers were implanted above the LDT of VGAT-Cre mice. In the control group, shVglut2-mCherry was replaced with shCtrl-mCherry (Fig. 7a). Four weeks after surgery, glutamatergic neurons in the LDT were

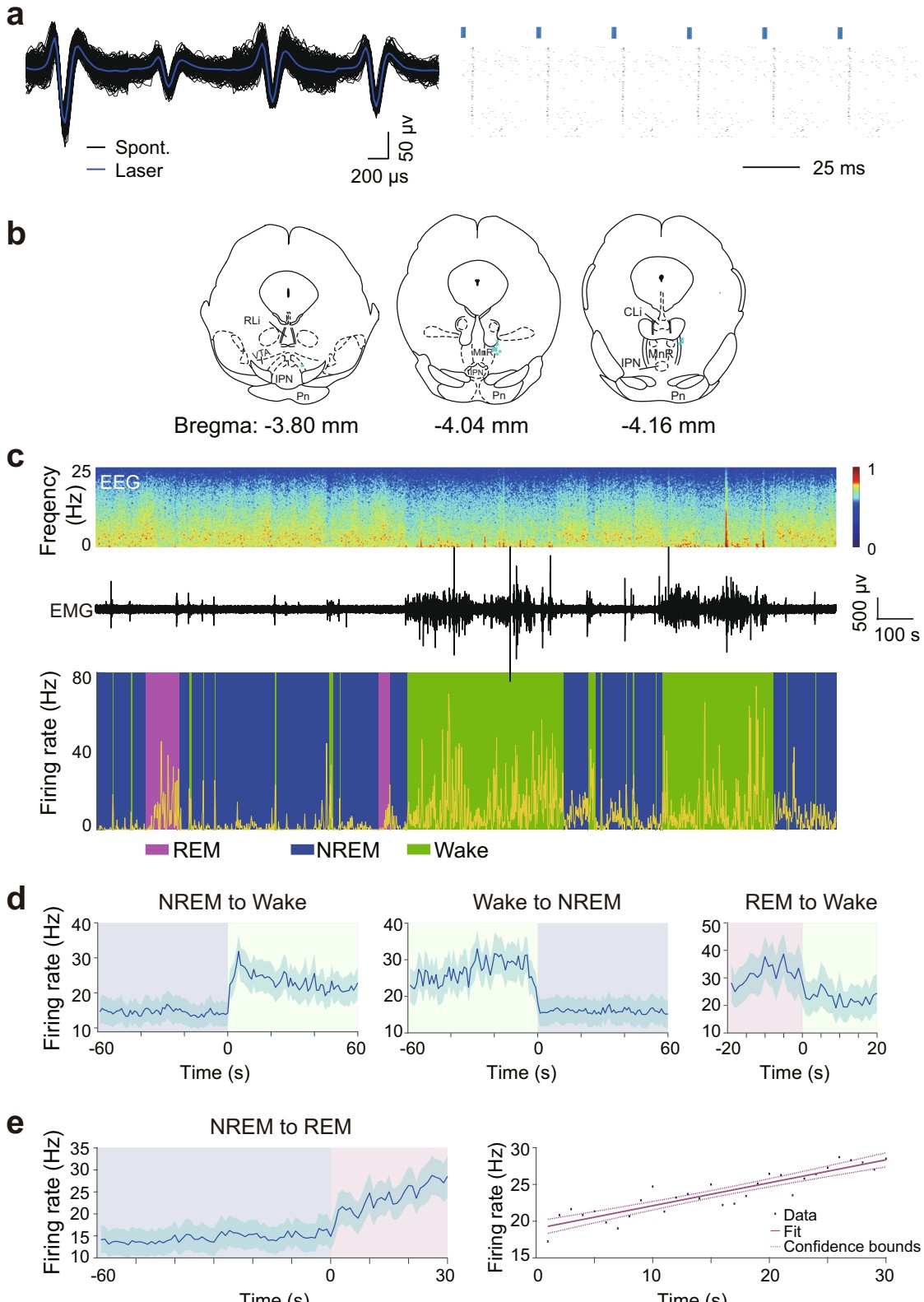

**Fig. 4 | Firing rates of identified RMTg GABAergic neurons across brain states.** **a** An example unit. Comparison between laser-evoked (blue) and spontaneous (black) spike waveforms from this unit (left). Spike raster showing multiple trials of laser stimulation at 30 Hz. Blue ticks, laser pulses (right). **b** Recording sites of 12 identified units across the rostrocaudal extent of the RMTg in VGAT-Cre mice. Each indigo dot indicates one unit. **c** Firing rates of an example RMTg GABAergic neuron (yellow) together with the EEG spectrogram, EMG amplitude, and color-marked brain state. **d** Mean firing rates of all identified GABAergic neurons at sleep–wake state transitions. **e** Firing rate changes of identified RMTg GABAergic neurons from NREM to REM sleep transition (left) and linear regression of firing frequency of all identified units during the initial 30 s of REM sleep (right). Dashed line: confidence bounds ($R^2 = 0.819$, $p < 0.01$). Shading indicates 95% confidence intervals (**d**, **e**). $n = 12$ units from 3 mice. Source data are provided as a Source Data file.

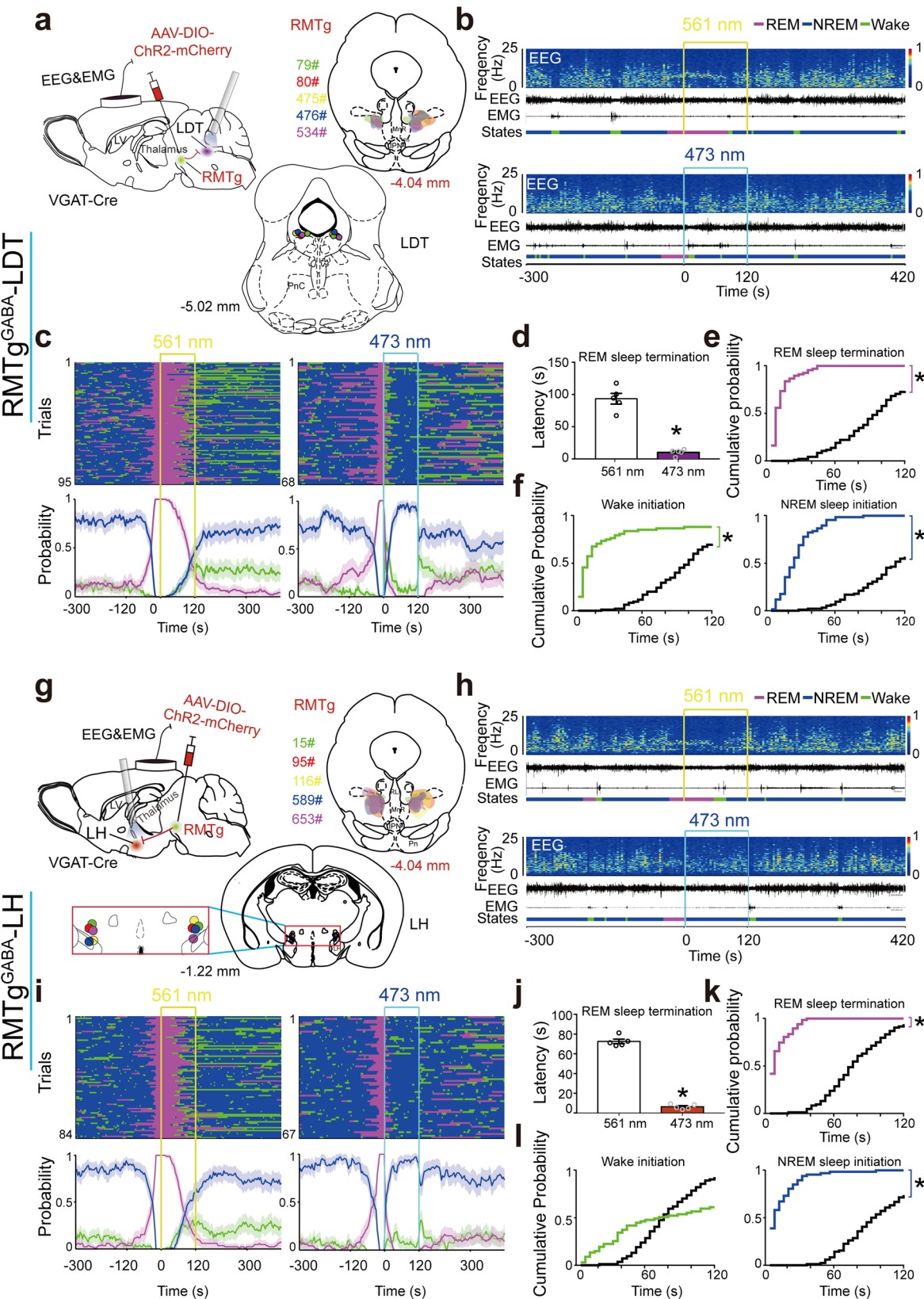

dramatically diminished in the shVglut2 group (Fig. 7b). Optogenetic activation of the RMTg GABAergic terminals in the LDT allowed for the transitions from REM to NREM sleep in mice with ablated glutamatergic neurons; meanwhile, REM sleep transitioned to arousal in control mice microinjected with shCtrl in the LDT (Fig. 7c, d).

Given that in vivo optogenetic activation of the RMTg GABAergic nerve endings in the LH converted REM to NREM sleep, we investigated the functional connections between RMTg GABAergic neurons and the different types of neurons in the LH. In vitro patch–clamp recordings showed that RMTg GABAergic neurons synapsed directly

**Fig. 5 | The RMTg^GABA^-LDT and RMTg^GABA^-LH circuits mediated distinct REM sleep transitions. a**, **g** Diagram of experiments for in vivo optogenetic stimulation of RMTg GABAergic terminals in the laterodorsal tegmentum (LDT) (**a**) or lateral hypothalamus (LH) (**g**). Schematic illustration of adeno-associated virus (AAV) encoding channelrhodopsin 2 (ChR2) fused to an enhanced red fluorescent protein mCherry delivered into the bilateral RMTg in VGAT-Cre mice (top left). The coronal section shows the superimposed virus-injected area (top right) and tips of optical fibers (bottom). **b**, **h** Representative EEG and EMG traces and corresponding heat map of the EEG power spectrum in different sleep–wake states induced by laser stimulation of the RMTg^GABA^-LDT (**b**) and RMTg^GABA^-LH (**h**). Top, control group with 561 nm laser stimulation. Bottom, 473 nm laser stimulation. **c**, **i** Sleep–wake state changes (top) and probability of brain state transitions (bottom) after photostimulation of RMTg GABAergic axons in the LDT (**c**) or LH (**i**) during REM sleep lasting no less than 16 s before laser on in all trials from

mice in each of these two circuit groups with 561 nm laser stimulation as the control (left) and 473 nm laser stimulation (right), respectively. Shading indicates 95% confidence intervals. Yellow and blue lines, 561 nm and 473 nm laser stimulation (30 Hz, 120 s, 10 ms pulse), respectively (**b**, **c**, **h**, **i**). **d**–**f**, **j**–**l** During 120 s laser stimulation of the RMTg^GABA^-LDT (**d**–**f**) and RMTg^GABA^-LH (**j**–**l**), mean latency of REM sleep transitions to wakefulness or NREM sleep after blue laser stimulation during REM sleep in the first eight trials per mouse. $p = 0.0312$ vs. control group, Wilcoxon test (**d**, **j**); cumulative probability of REM sleep termination, wake and NREM sleep initiation. *$p < 0.05$ for comparison between 473 nm laser stimulation and 561 nm control group, Kolmogorov–Smirnov test. Colorful stairs, group with 473 nm laser illumination. Black stairs, control group with 561 nm laser illumination. Data represent mean ± SEM. $n = 5$ for the circuits of RMTg^GABA^-LDT or RMTg^GABA^-LH. Source data are provided as a Source Data file.

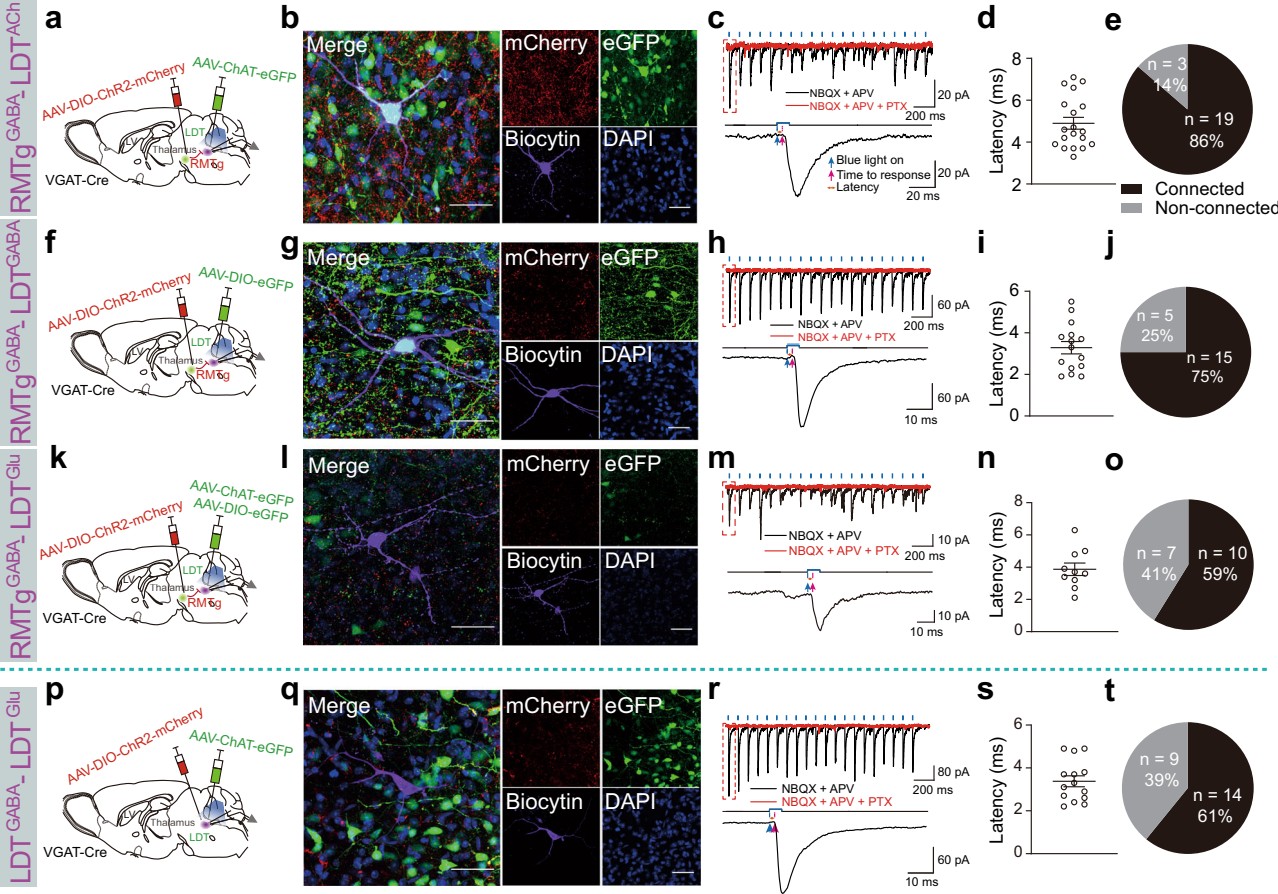

**Fig. 6 | RMTg GABAergic neurons directly inhibited LDT cholinergic, GABAergic, and glutamatergic neurons and indirectly disinhibited LDT glutamatergic neurons. a–o** RMTg GABAergic terminals formed inhibitory connections with laterodorsal tegmentum (LDT) cholinergic, GABAergic, and glutamatergic neurons. **p–t** Within the LDT, GABAergic neurons formed inhibitory connections with glutamatergic neurons. **a**, **f**, **k**, **p** Schematic diagrams of the experimental protocols. In VGAT-Cre mice, Cre-dependent AAVs encoding ChR2 was injected into the RMTg (**a**, **f**, **k**) or the LDT (**p**), whereas AAVs encoding the promoter of cholinergic neurons (ChAT-eGFP) (**a**), DIO-eGFP (**f**), ChAT-eGFP and DIO-eGFP (**k**), or ChAT-eGFP (**p**) were injected into the LDT of VGAT-Cre mice. **b**, **g**, **l**, **q** Representative images showing a recorded biocytin-filled neuron (violet) that is a cholinergic neuron (ChAT-eGFP: green) (**b**), GABAergic neuron (DIO-eGFP: green) (**g**), a glutamatergic neuron with no eGFP expression (**l**), and glutamatergic neuron with no eGFP or mCherry expression (**q**). Scale bar: 50 μm. **c**, **h**, **m**, **r** Typical traces of inhibitory postsynaptic currents (IPSCs) from an LDT cholinergic neuron

(**c**), GABAergic neuron (**h**), and glutamatergic neuron (**m**, **r**) evoked by blue light stimulation (black line) of ChR2-expressing GABAergic terminals in the RMTg (**c**, **h**, **m**) or LDT (**r**). The evoked currents were completely abolished by the application of picrotoxin (PTX) (red line) but not NBQX and APV (black line). **d**, **i**, **n**, **s** Latency of light-evoked IPSCs in LDT cholinergic neurons (**d**), GABAergic neurons (**i**), and glutamatergic neurons (**n**, **s**) that responded to laser stimulation of ChR2-expressing GABAergic terminals in the RMTg (**d**, **i**, **n**) or LDT (**s**). Data represent mean ± SEM. **e**, **j**, **o**, **t** Proportion of recorded LDT cholinergic (**e**), GABAergic (**j**), and glutamatergic neurons (**o**, **t**) that responded to laser stimulation of ChR2-expressing GABAergic terminals in the RMTg (**e**, **j**, **o**) or LDT (**t**). $N = 22$ cells (**e**), $N = 20$ cells (**j**), $N = 17$ cells (**o**), and $N = 23$ cells (**t**) from 3 mice. NBQX 6-nitro-7-ulphamoylbenzo(f)-quinoxaline-2,3-dione, APV d-(·)-2-amino-5-phosphonopentanoic acid, ACh cholinergic neurons, Glu glutamatergic neurons. Source data are provided as a Source Data file.

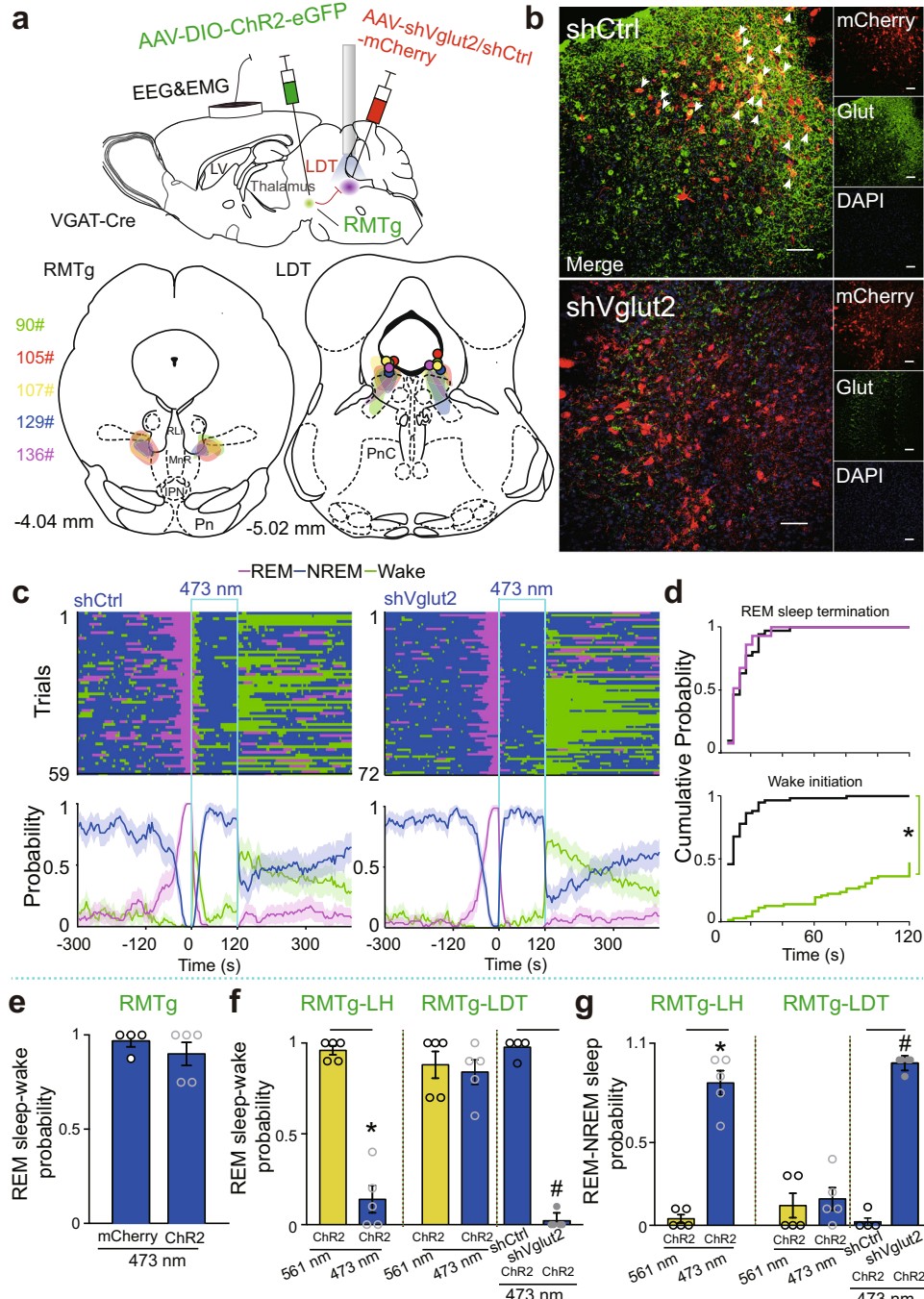

**Fig. 7 | Causal evidence for transient arousal from REM sleep by activation of the RMTg$^{GABA}$-LDT circuit. a** Schematic diagram of the experimental protocol (top). The coronal section shows the superimposed virus-injected area (bottom, left: RMTg; right: LDT) and the tips of optical fibers in the LDT (bottom, right). **b** Representative photographs of coronal sections containing the LDT from a control (top) and a knockdown (bottom) VGAT-Cre mouse. The images show mCherry-expressing neurons (red), glutamate (Glut)-expressing neurons (green, but not the green ChR2-expressing RMTg GABAergic nerve endings), merged neurons (yellow) pointed to by arrows with Glut and mCherry, and 4′,6-diamidino-2-phenylindole (DAPI, blue) staining. Scale bar, 50 μm. **c** Sleep–wake state changes (top) and probability of brain state transitions (bottom) after photostimulation of RMTg GABAergic axon endings in the LDT during REM sleep lasting no less than 16 s before laser on in all trials from mice in the shCtrl (left) and shVglut2 (right) groups. Shading indicates 95% confidence intervals. **d** During 120 s laser stimulation of the

RMTg$^{GABA}$-LDT, the cumulative probability for REM sleep termination and wake initiation. Colorful stairs, shVglut2 group. Black stairs, shCtrl group. *$p < 0.05$, Kolmogorov–Smirnov test. **e** Probability of first transitions from REM sleep to wakefulness by ChR2-mediated activation of RMTg GABAergic neurons. ChR2: $n = 5$, mCherry: $n = 4$. **f**, **g** Comparison between the probability of first transitions from REM sleep to wakefulness (**f**) and REM to NREM sleep (**g**) by activation of the RMTg$^{GABA}$-LH ($n = 5$), RMTg$^{GABA}$-LDT ($n = 5$), and RMTg$^{GABA}$-LDT with shVglut2 ($n = 5$) and shCtrl ($n = 4$) microinjection in the LDT, respectively. A 561 nm yellow laser stimulation was used as the control group. A 473 nm blue laser stimulation was used as the experimental group. *$p < 0.05$, Wilcoxon signed rank test for the experimental group with the 561 nm control group; #$p < 0.05$, Mann–Whitney $U$ test for the shVglut2 and shCtrl group. Data represent mean ± SEM. Source data are provided as a Source Data file. LDT laterodorsal tegmentum.

onto GABAergic or glutamatergic neurons in the LH (Fig. S13a–j). Orexin neurons in the LH also formed synaptic connections with RMTg GABAergic axons; whereas, light-evoked IPSCs were produced for longer than 5 ms, indicating indirect synaptic connections (Fig. S13k–o). Surprisingly, there were no synaptic connections between the RMTg GABAergic terminals and the LH melanin-concentrating hormone (MCH) neurons (Fig. S13p–s).

In summary, optogenetic activation of RMTg GABAergic neurons facilitated REM sleep transitions to wakefulness, with a dramatic decrease in latency (Fig. 2f). These findings are the same as those showing the activation of RMTg GABAergic terminals in the LDT, while stimulation of the terminals in the LH boosted REM transitions to NREM sleep. Ablation of LDT glutamatergic neurons prevented REM sleep–wake transitions and promoted REM–NREM sleep transitions when the RMTg GABAergic axons in the LDT were stimulated (Fig. 7e–g). These results revealed the necessary roles of LDT gluta-matergic neurons in the REM sleep-wakefulness transitions induced by ChR2-mediated stimulation of RMTg GABAergic neurons.

## Discussion

With the use of chemogenetics and optogenetics combined with optrode, EEG, and patch–clamp recordings, we revealed the role of a specific population of the RMTg in controlling REM sleep, constituting a key area in the REM sleep circuitry. First, we found that chemogenetic activation of RMTg GABAergic neurons caused fewer REM sleep epi-sodes, suggesting suppression of REM sleep initiation. To observe the immediate effects that follow the activation of RMTg GABAergic neu-rons during a specific brain state, we employed optogenetics, a widely applicable tool for providing a reliable and millisecond timescale control of neuronal activity[36]. Optogenetic activation of RMTg GABAergic neurons suppressed the initiation and maintenance of REM sleep when photostimulation was selectively given during NREM and REM sleep, respectively.

To detect the role of RMTg GABAergic neurons in the physiolo-gical regulation of REM sleep, we used chemogenetics to specifically inhibit the activity of these neurons. We found that REM sleep did not increase until after 2 h of strong and sustained wakefulness caused by chemogenetic inactivation of RMTg GABAergic neurons with CNO injection. The increased amount of REM sleep was mainly reflected by the prolonged length of the REM sleep segments. These findings were similar to those of our previous rat experiments, in which prolonged REM fragment durations were observed after RMTg neurons were lesioned with ibotenic acid[21]. However, after saline administration in mice expressing hM4Di in the RMTg GABAergic neurons, the sleep-deprived mice showed no rebound of REM sleep after 2 h of SD. These results are consistent with the findings that 3-h or 6-h SD through gentle handling in the early part of the light period in rats did not induce REM sleep enhancement[37]. Given that the biological effects of CNO could maintain for 6–10 h[38], the rapid and significant promotion of REM sleep compared with that in the SD group was due to the inactivation of RMTg GABAergic neurons rather than the rebound of REM sleep following sleep insufficiency. Considering that lesion for-mation causes irreversible damage to neurons and fibers of passage[39] and that chemogenetics lacks temporal precision, we explored the stage transitions induced by the inactivation of RMTg GABAergic neurons using optogenetics. We found that NREM sleep episodes were completely and promptly converted to REM sleep, and the duration of REM sleep episodes was prolonged by the Arch-mediated inactivation of RMTg GABAergic neurons. These findings clearly demonstrate that RMTg GABAergic neurons suppress REM sleep by preventing its initiation and promoting its termination.

Similar to NREM sleep, REM sleep is under homeostatic control. Following deprivation, the amount of REM sleep increases above the baseline level[40,41]. REM sleep is triggered by sufficient REM sleep pressure, which accumulates during NREM sleep and dissipates during REM sleep. In the mutual inhibition model for REM sleep regulation, REM sleep pressure is assumed to exert inhibitory effects on the excitability of REM-off neurons; thus, the firing rates of REM sleep neurons progressively increase and reach a higher level at the end of REM sleep[42–44]. In the present study, the RMTg GABAergic neuron activity was lower at the onset of REM sleep and gradually increased to reach relatively high levels at the termination of REM sleep. These findings are consistent with the temporal profile of vlPAG GABAergic and DR serotonergic neurons, which are known as REM-off neurons[14,45]. Additionally, our data are consistent with the reciprocal interaction model proposed by Robert McCarley and Allan Hobson, in which REM-on cholinergic neurons excite REM-off neurons during REM sleep, ultimately leading to the termination of REM sleep[46]. Therefore, the discharge activity changes across sleep–wake states and the correla-tions with REM sleep revealed by optrode recordings support the role of RMTg GABAergic neurons in suppressing REM sleep in behavioral experiments in mice.

Wake-promoting neurons, including dopaminergic neurons in the VTA and GABAergic neurons in the ventral pallidum (VP), increased the population activity to a higher level prior to the transition from NREM sleep to wakefulness[31,47]. However, in RMTg GABAergic neurons, there was an increase in firing rates about 30 s before the wake–NREM sleep transitions. The finding that the RMTg GABAergic neurons had high firing rates during the awake state but promoted NREM sleep when activated with optogenetics is in line with the NREM sleep promotion of vlPAG and VTA GABAergic neurons[14,48]. Importantly, RMTg GABAergic neurons had a distinct discharge pattern of phasic firing, in which spiking frequency peaks corresponded to EEG delta oscillations (Fig. S10), which may contribute to NREM sleep promotion. Considering that dopaminergic neurons in the VTA and GABAergic neurons in the VP are present with high levels of neuronal activity during wakefulness, the high firing rate of RMTg GABAergic neurons during wakefulness may protect the brain from over-activation by wake-active neurons.

A large body of evidence has revealed that LDT cholinergic neu-rons play an important role in the generation of REM sleep. Similarly, the majority of LDT GABAergic and glutamatergic neurons are more active during REM sleep and wakefulness[33–35]. The present in vitro electrophysiological recordings illustrated that RMTg GABAergic neurons directly synapse onto LDT cholinergic, GABAergic, and glu-tamatergic neurons. Moreover, within the LDT, we found that LDT GABAergic neurons form inhibitory connections with the local gluta-matergic neurons. Thus, inhibitory inputs from RMTg GABAergic neurons potentially exert both inhibitory and disinhibitory effects on LDT glutamatergic neurons. After the LDT glutamatergic transmission was selectively interrupted through RNA interference in mice, REM sleep was still rapidly terminated after activation of the RMTg^GABA-LDT circuit, which promoted the conversions from REM sleep to NREM sleep rather than to wakefulness. These findings suggest that RMTg GABAergic neurons may suppress REM sleep through direct inhibition of LDT cholinergic or GABAergic neurons and mediate the transition from REM sleep to an awake state through the disinhibition of LDT glutamatergic neurons (Fig. 8).

The LH coordinates various fundamental behaviors such as sleeping, waking, feeding, and motivated behaviors[49]. There are mul-tiple cell types in the LH, including GABAergic, glutamatergic, MCH, and orexin neurons[50]. GABAergic LH neurons have been reported to promote REM sleep by inhibiting noradrenergic neurons in the locus coeruleus (LC) or brainstem vlPAG/deep mesencephalic reticular nucleus GABAergic neurons[51–54], and these play an important role in promoting wakefulness as well[55]. In addition to GABAergic neurons, the LH also contains a large number of glutamatergic neurons that are active during REM sleep and wakefulness. Activation of these neurons induces and maintains arousal[56]. In the LH, neuropeptidergic neurons expressing the neuropeptides MCH and orexin, each of which only comprises a small population of LH neurons[49], have been shown to

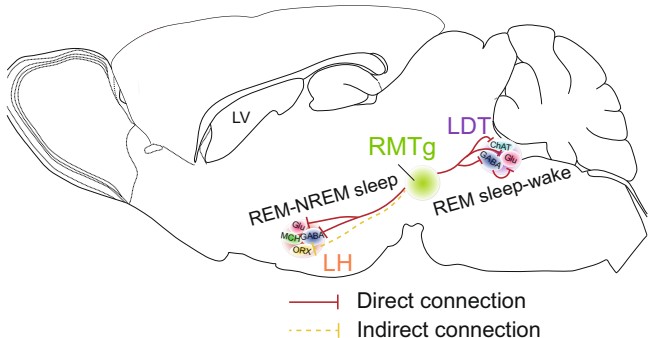

**Fig. 8 | Hypothesis for neural mechanisms of RMTg GABAergic neurons suppressing REM sleep through the LDT/LH.** RMTg GABAergic neurons inhibited cholinergic and/or GABAergic neurons in the LDT to cease REM sleep. Activation of RMTg GABAergic neurons inhibited GABAergic interneurons in the LDT, which then disinhibited glutamatergic neurons in the LDT to facilitate REM sleep–wake transitions. Activation of RMTg GABAergic terminals in the LH facilitated REM–NREM sleep transitions through direct inhibition of glutamatergic and/or GABAergic neurons and/or through indirect synaptic connection with orexin neurons. RMTg rostromedial tegmental nucleus, LDT laterodorsal tegmentum, LH lateral hypothalamus, LV lateral ventricle, ChAT cholinergic neurons, Glu glutamatergic neurons, ORX orexin neurons, MCH melanin-concentrating hormone neurons.

promote REM sleep[57] and prolong REM sleep episodes[58], respectively. However, the role of orexin neurons in the regulation of REM sleep remains controversial. Overexpression of orexin peptides in transgenic mice or intracerebroventricular administration of orexin results in reduced REM sleep[59–61]. Additionally, orexin neurons play a major role in the maintenance of arousal and increase the transitions to wakefulness[62,63]. Given the findings from patch-clamp recordings that RMTg GABAergic axons form inhibitory synaptic connections with GABAergic and glutamatergic neurons but not with MCH neurons in the LH, it is likely that activation of the RMTg[GABA]-LH circuit may suppress REM sleep and facilitate NREM sleep through direct inhibition of LH GABAergic and/or glutamatergic neurons or possibly through the indirect inhibition of LH orexin neurons (Fig. 8).

In addition to the different types of neurons in the LDT and LH, GABAergic neurons in the DR, another RMTg[GABA]-projecting target, have been reported to inhibit serotonergic neurons through local networks[64,65]. Taken together, these findings suggest that inhibiting LDT cholinergic/GABAergic and/or LH GABAergic/glutamatergic/orexin and/or disinhibiting DR serotonergic neurons may all be implicated in suppressing REM sleep induced by RMTg GABAergic neurons. Further studies are needed to uncover the precise neuronal circuits of RMTg GABAergic neurons in suppressing the onset and maintenance of REM sleep[66].

All species need sleep; however, sleep state transitions are distinct[67]. In rodents, transitions between REM and subsequent NREM sleep episodes, in most cases, include brief awake interruptions that are not common in humans. Waking up from sleep helps rodents stay alert to their surroundings to protect them from predators and natural hazards[68–71]. Two recent studies showed that the activation of basal forebrain parvalbumin neurons or LH GABAergic neurons can wake animals from NREM sleep but not from REM sleep[53,72], which requires a higher arousal threshold[73]. Interestingly, we identified a population of RMTg GABAergic neurons that quickly and selectively arouse mice at REM sleep by disinhibiting glutamatergic LDT neurons. A recent study revealed a similar finding, in which activation of the subthalamic corticotropin-releasing hormone neurons mediates waking from REM sleep in response to predator odors[74]. These findings further our understanding of how the brain controls arousal at different sleep stages when animals are challenged with dangerous situations or biologically relevant stimuli. However, increased arousal in humans

can prevent restorative sleep and contribute to physiological dysfunctions and pathophysiological mechanisms. For instance, recurrent trauma-related nightmares and frequent awakenings are highly prevalent in patients with PTSD[7,8]. Given that dreams most commonly occur during REM sleep[75], recurrent awakening from nightmares may be related to an increase in REM sleep in these patients[76]. In the present study, we found that silencing LDT glutamatergic cells during the activation of the RMTg[GABA]-LDT circuit or activation of the RMTg[GABA]-LH pathway substantially promoted REM to NREM transitions. These findings are of potential clinical significance for people with affective disorders who suffer from unwanted arousal from sleep, particularly REM sleep. In conclusion, this study shows that RMTg GABAergic neurons gate REM sleep through inhibitory inputs to the LDT and LH.

Sleep is not only regulated by homeostasis and circadian factors, but is also influenced by a variety of emotional and physiological factors, such as stress, pain, eating, and body temperature[77]. Using cell-type-specific retrograde trans-synaptic rabies virus tracing techniques, we found that RMTg GABAergic neurons receive wide projections in the brain, including dense projections from the LH, zona incerta, and LHb. Moreover, several brain regions, such as the PAG, DR, and LC, which contain a large number of REM-off neurons, also send inputs to RMTg GABAergic neurons[3,78,79]. Therefore, more research is needed on which emotional or physiological factors or which input circuit is implicated in the activation of RMTg GABAergic neurons to regulate sleep.

## Methods
### Ethics statement
All experimental procedures were approved by the Committee on the Ethics of Animal Experiments of the School of Basic Medical Sciences, Fudan University (license identification number: 20210302-105). The animals were anesthetized with 1.5–2% isoflurane before surgery or euthanasia. The animals were placed on a heating pad until they woke up from anesthesia after surgery for electrode implantation, AAV microinjection, or optical fiber implantation. The animals were observed daily, and their housing sawdust was kept clean during the postoperative recovery periods. Every effort was made to minimize the number of animals used or any pain or discomfort in the animals.

### Animals
Chemogenetic and optogenetic experiments and in vivo electrophysiological recordings were performed in male VGAT-Cre mice (Jackson Laboratory stock 017535, 8–10 weeks old)[80]. Both male and female VGAT-Cre mice were used in the anterograde tracing experiments (8–10 weeks old) and in the in vitro electrophysiological recordings (4–6 weeks old). Mice were housed at an ambient temperature (22 ± 0.5 °C) with a relative humidity of 60% ± 2% and a 12-h light/dark cycle (lights on at 07:00, illumination intensity of approximately 100 lx). Food and water were provided *ad libitum*.

### Virus preparation
Viruses AAV2/9-hSyn-DIO-ChR2-mCherry-WPRE-pA, AAV2/9-hSyn-DIO-mCherry-WPRE-pA, AAV2/9-hSyn-DIO-ArchT-eGFP-WPRE-pA, AAV2/9-hEF1a-DIO-hChR2(H134R)-eGFP-WPRE-pA, AAV2/9-hSyn-hM3Dq-mCherry-WPRE-pA, AAV2/9-hSyn-hM4Di-mCherry-WPRE-pA, and AAV-hSyn-DIO-eGFP were purchased from Taitool. rAAV-ChAT-eGFP-WPRE-pA was purchased from BrainVTA[81]. AAV-MCH-eGFP and AAV-orexin-eGFP were designed and produced by BrainCase and BrainVTA, respectively, and tested in this study (Fig. S14). AAV-shVglut2-mCherry and AAV-shCtrl-mCherry were provided by Professor Michael Lazarus at the University of Tsukuba, Japan.

### Surgery
After anesthesia, mice aged 8–10 weeks were placed on a stereotaxic apparatus. After asepsis, the skin was cut to expose the skull, and the

overlying connective tissue was removed. A small craniotomy was performed above the superficial layer of the RMTg or the LDT. The coordinates for the AAV injections were as follows: RMTg: anteroposterior (AP) − 3.7 mm, mediolateral (ML) 0.5 mm, dorsoventral (DV) 4.0 mm. LDT: AP − 4.7 mm, ML 0.5 mm, DV 2.5 mm. The AAV vectors were slowly injected (40 nL/min) into the bilateral RMTg (30 nL on each side) or LDT (70 nL on each side).

For in vivo optogenetic manipulation, additional bilateral optical fibers (numerical aperture, 0.37) were implanted above the targeted nuclei, including the RMTg, LDT, and LH. One optical fiber was implanted vertically and the other was implanted ten degrees to the vertical line. The locations of the screws were adjusted accordingly. Coordinates for optical fiber implantation: RMTg: AP − 3.7 mm, ML 0.5 mm, DV 3.7 mm; AP − 3.7 mm, ML 1.2 mm, DV 3.7 mm, $\theta = 10°$ for contralateral implantation; LDT: AP − 4.7 mm, ML 0.5 mm, DV 2.0 mm; AP − 4.7 mm, ML 0.85 mm, DV 2.0 mm, $\theta = 10°$ for contralateral implantation; LH: AP − 1.3 to 1.5 mm, ML 1.0 mm, DV 4.2 mm; AP − 1.3 to 1.5 mm, ML 1.8 mm, DV 4.2 mm, $\theta = 10°$ for contralateral implantation. For in vitro electrophysiological experiments, mice aged 4–6 weeks were used, and the coordinates for AAV injections were as follows: RMTg: AP − 3.5 mm, ML 0.5 mm, DV 4.0 mm; LDT: AP − 4.5 mm, ML 0.5 mm, DV 2.5 mm; LH: AP − 0.6 to −0.8 mm, ML 1.0 mm, DV 4.8 mm.

Two stainless steel screws of EEG electrodes were inserted into the skull 1.5 mm from the midline, 1.0 mm anterior to the bregma, and 3.0 mm posterior to the bregma. Two EMG cables were inserted into the trapezius muscle. Finally, the electrodes were immobilized using dental cement and attached to the skull. For chemogenetic experiments, electrodes for recording EEG and EMG signals were implanted three weeks after virus injection. For in vivo optogenetic experiments, in addition to electrodes for recording EEG and EMG signals, optical fibers were also implanted three weeks after virus injection. For in vivo electrophysiological recordings, a self-designed and self-made optrode was implanted into the RMTg within 30 min. Simultaneously, a mini-electrode for EEG and EMG recordings was implanted into the skull, and the screws were adjusted accordingly. One to two weeks after the surgery, in vivo data were collected. Mice housed three to four mice per cage, except for mice with implanted optical fibers or mice used for spike recordings, which were housed individually. Only data from mice in which the sites of viral infections and optical fiber implantation were confirmed were accepted[21].

## Sleep recording and vigilance state analysis
After 2 weeks of recovery from the EEG/EMG electrode implantation, the mice were transferred to the recording room and housed individually in transparent barrels for habituation. The electrodes of each mouse were linked to a cable that was connected to a slip ring. The mice were habituated to the recording cables and conditions for at least three days. Following habituation, in the chemogenetic experiments, saline or CNO (C4759, LKT Laboratory, Minneapolis, USA, 1 mg/kg) dissolved in saline was injected intraperitoneally (i.p.) at 09:00 on days 1 and 2 at a volume of 10 ml/kg.

Cortical EEG and EMG signals were amplified and filtered (EEG, 0.5–30 Hz; EMG, 20–200 Hz) at a sampling rate of 128 Hz and recorded using VitalRecorder (Kissei Comtec, Nagano, Japan). When complete, the polysomnographic recordings were automatically scored offline every 4 s as wakefulness, NREM sleep, or REM sleep using SleepSign 3.0, based on the published standard criteria. Defined sleep–wake stages were manually corrected, if necessary. The investigator, who checked the EEG signals to determine the brain states, was blinded to group information[82].

In the optogenetic experiments, the optical fiber was connected to a laser stimulator (ADR-700A, Shanghai, China) through an optical cable. The stimulation parameter was set using Master 9 (AMPI, Israel), which simultaneously connects to the laser stimulator and the digital-to-analog converter of the computer. For optogenetic activation, a

473 nm laser (2–5 mW/mm$^2$ at the fiber tip) was used for 120 s with a pulse duration of 10 ms at 30 Hz. For optogenetic inactivation, 561 nm constant laser illumination (2–5 mW at the fiber tip) was used for 60 s based on the literature[12,83]. The optogenetic experiments on sleep–wake behaviors were performed during both the light (7:00–19:00) and dark period (19:00–7:00). A laser was randomly delivered to different brain states of REM sleep, NREM sleep, or wakefulness with an interval of 10 min, given that REM sleep occurs every 10–20 min in mice[77]. EEG signals were scored offline using SleepSign 3.0 to determine different vigilance states. All trials in which light stimulation was initiated when mice naturally entered REM sleep for at least 16 s or NREM sleep/wakefulness for at least 60 s were used for analysis[84]. The cumulative initiation probability was calculated as the ratio of cumulative trials entering a specific brain state during laser stimulation to the total trials at each time point of stimulation. The cumulative termination probability was calculated as the cumulative trials converting to the other two brain states divided by all trials at each time point of stimulation[14].

## In vivo electrophysiological recordings
We used modified tetrodes or optetrodes for the electrophysiological recordings[85]. Four nickel–cadmium wires were fused together (350 °C) as a tetrode, except for the side (1 cm long), which was kept separate for welding to the connector. Four pairs of tetrodes were guided by silica tubes that were fixed to the screw nut. Nickel–cadmium wires (R0800, KANTHAL Precision Technology, Hazelwood, USA), silica tubes (TSP075150 Polymicro Technologies, USA), and a screw with screw nut were placed in a resinous frame (designed using SolidWorks 2018 software and printed using Wenext, Shenzhen, China). The inner wall of the frame and the screw nut were adjoined so that the screw nut would not rotate while the screw was turned. The design of the frame allowed the tetrodes to be adjusted for further in vivo recordings. Each of the 16 wire tips was welded to the corresponding pins on a PCB (printed circuit board) connector that was electrically connected to an adapter. Two electrodes for grounding and reference were welded directly to the corresponding connector pins of the Omnetics adapter (Omnetics Connector Corporation, USA), which were then glued to the external wall of the resinous frame. For the optetrodes, an optical fiber was fixed to the frame directly adjacent to the tetrodes within 0.5 mm. Finally, the tips of the tetrodes were trimmed and aligned with the optical fiber. The tetrodes were electroplated to a final impedance of approximately 0.5 MΩ, with an automatic multichannel electroplating system (ADPT-nanoZ-NN-16, multichannel systems, USA; Neuralynx, USA). A gold solution was prepared for electroplating (Japan). Wires to record the cortical EEG and EMG signals were also attached for simultaneous recording. The recording optetrodes were lowered by 55 μm each day by turning the screw by approximately 90°.

The surgical optetrode implantation was similar to the optogenetic implantation. The body temperatures of mice were maintained throughout the procedure using a heating pad. After asepsis, the skull (1 mm × 1 mm) and dura were gently removed above the superficial layer of the RMTg. We then cleaned the brain tissue and slowly implanted the optetrodes (DV 3.9 mm). One stainless steel screw was inserted into the skull (AP 1.0 mm) and two stainless steel screws were inserted into the skull (AP −5.5 mm) for welding to the ground and reference. After protection of the brain tissue out of the skull with tissue glue (KWIK-SIL, World Precision Instruments, USA), the electrodes were immobilized to the skull using dental cement. Finally, the optetrodes were enclosed in a copper network.

The mice recovered for two weeks prior to the recording with a 16-channel Plexon system and Omniplex software (Plexon, USA). Wideband signals (0.5–8000 Hz) from all tetrodes were amplified and digitized at 40 kHz for offline sorting. The EEG and EMG signals were recorded simultaneously at 200 Hz via an analog channel. Offline spike detection was performed using Offline Sorter V4. Raw data were

filtered using low cuts to remove field potentials. Typically, a negative value four times the standard deviation of the recorded signals of each lid of a tetrode is set as the threshold for detecting spike events. Single units were defined based on the waveform energy and the first three principal components of the spike waveforms on each lid of a tetrode. A single unit was included only if the firing rate exceeded 1 Hz. For optogenetic tagging, laser pulse trains (30 Hz, 30 s) were applied. A unit was identified as expressing ChR2 if spikes were reliably evoked by laser pulses with a short latency (<5 ms), as well as if the waveforms of the laser-evoked and spontaneous spikes were highly similar. The timestamp spiking activity was analyzed using NeuroExplorer 5 and MATLAB R2018b.

### In vitro electrophysiological recordings

To test the functional connections of RMTg GABAergic neurons to their targeted cells or the manipulation of the activity of the AAV-infected RMTg GABAergic neurons by optogenetic illumination or CNO administration, VGAT-Cre mice were bilaterally microinjected with AAV vectors carrying Cre-dependent hSyn-ChR2 (Arch, hM3Dq, hM4Di)-mCherry into the RMTg. After recovering for 3–4 weeks, the mice were anesthetized and transcardially perfused with ice-cold modified artificial cerebral spinal fluid (ACSF) saturated with 95% $O_2$ and 5% $CO_2$ and containing the following (in mM): 215 sucrose, 26 $NaHCO_3$, 10 glucose, 3 $MgSO_4$, 2.5 KCl, 1.25 $NaH_2PO_4$, 0.6 Na-pyruvate, 0.4 ascorbic acid, and 0.1 $CaCl_2$. The brains were rapidly removed, and acute coronal slices (300 μm) containing the RMTg, LDT, or LH were cut on a vibratome (VT1200, Leica, Heidelberger, Germany) in an ice-water mixture of modified ACSF with 95% $O_2$ and 5% $CO_2$. The slices were then transferred to a holding chamber containing normal recording ACSF (in mM): 125 NaCl, 26 $NaHCO_3$, 25 glucose, 2.5 KCl, 2 $CaCl_2$, 1.25 $NaH_2PO_4$, and 1.0 $MgSO_4$. Slices were incubated for 30 min at 32 °C and equilibrated for at least 30 min at room temperature (22 °C) before recording. For recording, the slices were moved to the submersion recording chamber and continuously perfused with oxygenated ACSF at 32 °C and at a rate of 2 mL/min. Neurons in the RMTg, LDT, and LH were identified under visual guidance using infrared differential interference contrast video microscopy with a 40× water immersion objective lens (BX51WI, Olympus). Images were captured using an infrared-sensitive CCD camera (PHOTOMETRICS, Iris 9). Recording pipettes (5–8 MΩ) were filled with an internal solution containing the following (in mM): 105 potassium gluconate, 30 KCl, 10 phosphocreatine, 4 ATP-Mg, 0.3 EGTA, 0.3 GTP-Na, and 10 HEPES (pH 7.3, 285–300 mOsm). In some experiments, 0.1% biocytin (v/v, Sigma, USA) was added to the internal solution. Recordings were conducted in the whole-cell configuration using a Multiclamp 700 B amplifier (Axon Instruments, USA). Signals were filtered at 2 kHz and digitized at 10 kHz using DigiData 1440 A (Axon Instruments, USA). Data were acquired and analyzed using the pClamp10.3 software (Axon Instruments, USA).

Responses were evoked by light flashes delivered from a microscope-mounted 473 nm (1–100 Hz, 5 ms) or 561 nm (continuing 1.5 s) LED (Lumen Dynamics, Canada) through an objective lens directed onto the slice. The power of the LED was 3–5 mW. Photostimulation was controlled via analog outputs of DigiData 1440 A, enabling control over the duration and intensity. In voltage-clamp mode, the cells were held at −60 mV. When needed, 25 μM APV, 5 μM NBQX, and 50 μM PTX were added to block NMDA, AMPA, and GABA$_A$ receptors, respectively. Cells with Ra changes of over 20% were discarded.

To verify the chemogenetic manipulation of RMTg GABAergic neurons, we randomly recorded neurons expressing the mCherry reporter gene. In the current clamp mode, with a holding voltage of −60 mV, we first recorded the baseline level and then added 5 μM CNO to excite the hM3Dq-expressing neurons or inhibit the hM4Di-expressing neurons in the RMTg. The firing frequency and cell membrane potential of the recorded RMTg GABAergic neurons before and after CNO treatment were analyzed.

### Immunohistochemistry

Mice were deeply anesthetized and transcardially perfused with phosphate-buffered saline (PBS), followed by 4% paraformaldehyde in PBS. For fixation, the brains were kept overnight in 4% paraformaldehyde. Each brain sample was placed in 30% sucrose in PBS for 48 h. After embedding and freezing, each brain was cut into 30 μm coronal slices using a cryostat (CM1950, Leica, Heidelberger, Germany).

For immunostaining, the brain slices were washed three times with PBS and incubated with primary antibodies (Foxp1: rabbit anti-Foxp1, 1:20,000, ab16645, Abcam, USA; GABA: rabbit anti-GABA, 1:1000; PA5-32241, Invitrogen, USA; Orexin: mouse anti-orexin-A,1:600, sc-80263, Santa Cruz Biotechnology, USA; MCH: rabbit anti-melanin-concentrating-hormone, 1:1000, M8440, Sigma-Aldrich, USA; Glutamate: rabbit anti-glutamate, 1:1000, G6642, Sigma, USA; and Biocytin: 1:1000, S21374, Invitrogen, USA) dissolved in PBST (0.3% Triton X-100 in PBS) overnight at 4 °C. The next day, slices were washed with PBS and incubated with secondary antibodies (Donkey anti-rabbit/goat, 1:1000; Jackson ImmunoResearch, USA) for 2 h. Fluorescence images were captured using a confocal microscope (Nikon AIR-MP).

### Statistical analysis

Data analysis was performed using MATLAB R2018b and Prism 7. Firing frequency during sleep–wake stage transitions and the effects of laser stimulation on sleep-state transitions were calculated using 95% confidence intervals. The cumulative probability of state initiation/termination was calculated, followed by a two-sample Kolmogorov–Smirnov test. For the latency of the state transitions or transition probability, the first 8–10 trials of each mouse were included and analyzed using Mann–Whitney $U$ tests or Wilcoxon signed-rank tests. In all cases, statistical significance was set at $p < 0.05$. Two-way repeated ANOVA followed by a Sidak post hoc test was used when performing comparisons of the hourly amounts and number of bouts with different stage durations between the different conditions (saline or CNO) in the chemogenetic experiments.

### Reporting summary

Further information on research design is available in the Nature Portfolio Reporting Summary linked to this article.

## Data availability

The raw data have been deposited at: https://doi.org/10.6084/m9.figshare.21459780.v3. Source data are provided in this paper.

## Code availability

The codes that support the findings of the present study are available from the corresponding author upon request.

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

## Acknowledgements

We are grateful to Michael Lazarus (International institute for integrative sleep medicine, University of Tsukuba, Japan) for kindly providing virus vectors encoding shVglut2/shCtrl. We thank Chengyu Li (Center for Excellence in Brain Science and Intelligence Technology, Chinese Academy of Sciences) for kindly sharing the optrodes-making process. We also thank Hui Dong (Department of Pharmacology, School of Basic Medical Sciences, Fudan University) for technical support and help with optogenetic analysis. This study was supported in part by grants from the National Natural Science Foundation of China (32170983 and 81571296 to S.R.Y.; 82071491 to W.M.Q.), Natural Science Foundation of Shanghai Science and Technology Commission (21ZR1408000 to S.R.Y.), and the National Major Project of China Science and Technology Innovation 2030 for Brain Science and Brain-Inspired Technology (2021ZD0203400 to Z.L.H.). We would like to thank Editage for the English language editing.

## Author contributions

S.R.Y., Z.L.H., and W.M.Q. conceived and supervised the study. S.R.Y. and Y.N.Z. designed the experiments and wrote the paper. Y.N.Z., S.R.Y., and J.B.J. analyzed the data. Y.N.Z., S.R.Y., J.B.J., S.Y.T., Y.Z., and Z.K.C. acquired data. All authors discussed the paper.

## Competing interests

The authors declare no competing interests.
