## [Peer Review File · Nature Communications]

GABAergic neurons in the rostromedial tegmental nucleus are essential for rapid eye movement sleep suppressionREVIEWER COMMENTS

Reviewer #1 (Remarks to the Author):

This manuscript found that RMTg GABAergic neurons promote REM sleep to wake transitions through disinhibition of glutamatergic LDT neurons. The experiments are rigorous, the manuscript is well written, very organized and uses many approaches to address the role of GABAergic RMTg neurons in REM sleep control. It is of great interest to the sleep field as well as the broader neuroscience community. The comments below aim to help clarify the important results and interpretations of the manuscript.

Major comments:

- Excitatory M3Dq experiments were conducted during the light period (inactive period). Given that exciting the RMTg previously was shown to promote sleep, please postulate if doing this experiment during the dark period (active period) have given a different outcome?

-On page 7, lines 112-114, the efficacy of the chemogenetic techniques was established. However, there is no verification of their specificity in GABAergic neurons. Do the authors have histology with immunohistochemistry for GABA neurons to confirm that the stimulated neurons are GABAergic? i.e. Verifying the specificity of the vGAT-cre mouse line used.

- Page 7, Line 117: the CNO dose used in mice should be listed somewhere at least once near the beginning of the results. It is mentioned for the patch clamp CNO dose used but seems to be missing for the behavioral experiment results in vivo.

-On page 8, lines 120-126, the authors say that activation of RMTg caused fewer REM sleep periods, not shorter REM sleep periods (Figure 1). However, on page 16, lines 242-245, it is shown that inactivation of RMTg causes longer REM sleep periods, as opposed to more REM sleep periods (Figure S4). The authors then suggest that this second result indicates that RMTg GABAergic neurons inhibit REM sleep maintenance, but that is inconsistent with the first result. The second result is brought up in the Discussion (page 42, lines 596-597), but the first result's conflict with this needs to be addressed.

- Page 16, Line 242-245: Another possible interpretation of the inhibitory DREADDs data is that CNO led to increased wakefulness, but that the increased sleep drive (due to the experiment being carried out between 9:00-17:00/ light period) may have made even a mild 2 hour sleep deprivation period enough to lead to a minor sleep rebound?

-One of the claims of the manuscript is that RMTg GABAergic neurons disinhibit LDT glutamatergic neurons (e.g., page 38, lines 537-538). However, RMTg GABAergic neurons may also synapse directly onto LDT glutamatergic neurons, as they do onto the other two types of LDT neurons, potentially exerting both disinhibitory and inhibitory effects on LDT glutamatergic neurons. Figure 6 shows the direct synaptic connections of RMTg neurons and ChAT/vGAT neurons, but does not show the IPSCs resulting from RMTg synapses onto glutamatergic neurons. It would be helpful for their conclusions if the authors included this data.

-Please elaborate in the discussion the apparent paradox of the RMTg GABAergic neurons having high firing rates during wake but when activated with optogenetics promote NREM sleep.

-Pages 58-59, lines 932-942. Although the authors specify that wakefulness, REM, and non-REM sleep were scored offline using SleepSign 3.0, they do not describe how this is evaluated online for the optogenetic stimulation experiments. Furthermore, they specify that stimulation during REM periods happens ≥ 16 s after the start of REM (or ≥ 60 s for NREM), but not how the timing of the optogenetic stimulation was determined beyond that. The figures show that sometimes the stimulation occurred long after 16 s, but other times it was shorter. Please describe the criteria for online sleep scoring and stimulation timing.

Minor comments:

-Page 1, line 16. Please change disorder to disruption or disturbances. "REM sleep disorder" is not a defined diagnosis and sounds a lot like "REM sleep behavior disorder" which I don't think you are talking about.

- Page 13, Line 185: These optogenetic experiments used 120 seconds of stimulation to see changes in sleep states. The authors showed previous data above testing different frequencies of stimulation, did they also test other durations of stimulation time?

-On page 13, line 191, I think "ChR2" and "control" were accidentally swapped.

-On page 13, line 199, should "ms" be "s"?

- Page 19, Line 282: why was only 60 seconds laser stimulation used in inhibitory optogenetic experiments? Usually it is much harder to push a system using inhibition compared to excitation (120s used for ChR2 experiments).

-Likewise, on page 22, line 326-327, should "10 ms" be "10 s"? A 30 Hz stimulation applied every 10 ms doesn't make much sense.

-Page 24, line 356: did tracing with unilateral microinjections label the contralateral RMTg or ipsilateral side?

-Page 24, line 358: define abbreviations

-It is unclear based on page 26, line 382, if the authors are stimulating GABAergic cell bodies or just nerve endings. The schematic in Figs. 5A and 5G make it look like the latter, so please correct line 382 if needed.

-Page 26, line 384: what frequency and duration was laser stimulation?

-Page 36, line 507: with respect to the optogenetic terminal patch clamp experiments carried out in the LH, it would be useful to mention if GABAergic neurons make up a large proportion of LH neurons or is this a subpopulation that the authors were trying to look at?

-Page 55, line 866: It would improve the manuscript's scope if both sexes were used, given sex-related differences in sleep (e.g., see Dib et al., 2021, *Neurobiol Sleep Circadian Rhythms*). Otherwise, the Results section needs to be specific when presenting the in vivo results that they are from male mice only.

-Please add a line explaining why you didn't continue to investigate RMTg neurons projecting to midbrain dopamine regions like the VTA and SN

-Was there blinding during any of the analyses or experiments?

Figures:

-In Figure 1B (and other figures), please show more than one slice so readers can judge the extent of the anterior-posterior expression, maybe in supplementary figures.

-On Figure 1H, please include the units on the x-axis. Multiple t-tests were used for comparing groups at different time points but this seems like a situation for a two-way ANOVA? It's unclear how many mice were used for patch clamp experiments to confirm CNO activation of dreads receptors vs. behavioral M3Dq experiments. Please add the n for these experiments.

-In Figure 2E, there appears to be some amount of rebound REM sleep happening after REM sleep is

suppressed, relative to the same time period during the control conditions. Analysis of this effect could be interesting for determining the physiological function of these neurons.

-Figure 3C, this panel is slightly confusing visually because both example traces list -60mV at the start of the trace but one actually represents the membrane potential measured and one the changes in current. Maybe the labeling can be adjusted slightly.

-Figure 4B: Please adjust the spectrogram color bar so you can see the oscillation dynamics more clearly

-Figure 4 C-D: color scheme makes it hard to see the confidence intervals. Maybe darken these lines or use shading if possible?

-Figure S5: it would be useful to give estimates of A-P location in the brain for slices shown

-Figure S6, Page 32, Line 442: "Colorful stairs... mCherry group..." do you mean control laser (yellow) vs blue light instead of ChR2 vs mCherry control group?

-Figure 6: this figure is confusing because initially the panels A, F, K look identical but closer examination shows that A,F experiments are exciting RMTg terminals in the LDT while panel K is using the same viruses to eliminate GABAergic and cholinergic neurons in the LDT and instead record from glutamatergic LDT neurons? The Figure 6 legend text for panels K-O might need another sentence to clarify this part of the figure and distinguish it from the panels above it. Also, Figure 6 and corresponding section 6 in the Results (page 32) might be clearer if they use "GABAergic RMTg," "cholinergic LDT" and "GABAergic LDT" neuron labels were clarified in the text and figure legend.

-Figure 7 E-G: please clarify the labeling in the figure and legend text

Reviewer #2 (Remarks to the Author):

The paper by Zhao et al. investigates whether and how GABAergic neurons in the rostromedial tegmental nucleus (RMTg) are involved in the suppression of REM sleep. At first, the authors labeled the Foxp1 immunoreactive neurons and found that Foxp 1 was expressed at higher levels and strongly colocalized with GABA in the RMTg. Consequently, the authors used viral approaches to target and manipulate the RMTg GABAergic neurons. The authors showed that optogenetic activation of the RMTg GABAergic neurons immediately converted REM sleep to arousal and then initiated non-REM sleep. Likewise, chemogenetic activation of the RMTg GABAergic neurons decreased the REM sleep episodes. In contrast, optogenetic inactivation completely transitioned non-REM to REM sleep and prolonged REM sleep duration. The REM sleep control was interpreted to a disinhibition effect on the LDT glutamatergic neurons. The involvement of RMTg GABAergic neurons in REM was also demonstrated by their firing properties during the sleep-wakefulness transition.

The authors conducted many experiments to address these questions, utilizing state of the art genetic circuit dissection techniques and challenging experimental approaches. Considering their strong influence on brain states, the data shown are interesting and suggest a potential role for the RMTg GABAergic neurons in controlling REM sleep. However, I'm not completely convinced of main conclusions drawn from the current results because of the following concerns:

Major concerns:

(1) The RMTg GABAergic neurons receive major input from the lateral habenula, which conveys negative reward and motivation related information. Moreover, it has been demonstrated that the RMTg GABAergic neurons project to ventral tegmental area, dorsal raphe nucleus and locus

coeruleus, regulating diverse functions such as aversive and despair behavior, addiction and pain. Why the RMTg GABAergic neurons are required to be activated during sleep and subserve for the suppression of REM sleep? How the RMTg GABAergic neurons are activated during sleep? More contexts should be provided for supporting and interpreting their claims.

(2) Optogenetic and chemogenetic manipulation of the RMTg GABAergic neurons are used as a major argument for their causal role in REM sleep regulation. However, it is not mentioned how the authors controlled the location specificity of their virus injections. Because the RMTg is a rather elongated and irregular structure (see Fig. S1B), it is hard to know for sure that all RMTg and only RMTg are injected with virus. Indeed, as shown in Fig. 1B, one of its immediate neighbors, the VTA is obviously transfected by the virus, which thus weakens their claims on the role of RMTg GABAergic neurons in REM sleep control. Data strictly viral expressing in the RMTg for each tested mouse would be helpful to strengthen their claims.

(3) Firing rates of the RMTg GABAergic neurons at brain state transitions are used as another major argument for their role in REM sleep control. Based on their claims, the RMTg GABAergic neurons might function as REM-off units. However, the activity of RMTg GABAergic neurons significantly increased during the transition from NREM to REM sleep (Fig 4D). How to explain these contradictory results? Likewise, how to explain the decreased firing of RMTg GABAergic neurons during the wake-NREM transition (Fig 4C)? As shown in Fig. S3A, optogenetic activation of RMTg GABAergic neurons resulted in increased NREM sleep.

(4) Did all RMTg GABAergic neurons showed similar firing pattern during brain state transitions. The authors should make careful analysis for each recorded RMTg GABAergic neurons in Fig.4, and compared their firing activity across distinct states. This analysis is necessary because the RMTg GABAergic neurons project to diverse downstream targets and play various roles as recently reported.

(5) The downstream LH neurons for the RMTg GABAergic neurons should also be identified which is critical for thorough understanding their hypothesis listed in Fig. 8.

Minor points:

(1) Given the strong influence of optogenetic inactivation, it is hard to understand why chemogenetic inactivation of the RMTg GABAergic neurons only plays subtle role in initialize or stabilize REM sleep as shown in Fig S4D, E. More explanations should be made for these results.

(2) It is hard to understand the in vivo recording results listed in Fig 4A. Are those spikes recorded from 4 channels of an optrode, or 4 different RMTg GABAergic neurons? In addition, scale bar for this panel was incorrect. For example, 500 ms should be 500 μ m.

Reviewer #3 (Remarks to the Author):

Zhao et al. found that the rostromedial tegmental nucleus (RMTg) GABAergic (RMTg-GABA) neurons play an essential role in the regulation of rapid eye movement (REM) sleep. Chemogenetic activation of RMTg-GABA neurons decreased amount of REM sleep by decreasing the number of transitions from non-REM (NREM) sleep to REM sleep. Optogenetic activation of RMTg-GABA neurons during REM sleep induced transition to wakefulness (Wake) and then NREM sleep. Optogenetic inhibition of RMTg-GABA neurons during NREM sleep increased transition to REM sleep, and that during REM sleep increased the latency of REM sleep transition. In vivo electrophysiological recording of RMTg-GABA neurons revealed that those neurons are active during both Wake and REM sleep, while the activity was gradually increased during the first 30 sec of REM sleep. Additionally, optogenetic activation of nerve terminal of RMTg-GABA in the laterodorsal tegmental (LDT) nucleus induced transition to Wake and then NREM sleep, while that in the lateral hypothalamus (LH) induced direct transition to NREM sleep. In vitro optogenetic recording revealed that both cholinergic and GABAergic

neurons in the LDT receive inhibitory input from RMTg-GABA neurons, and glutamatergic neurons in the LDT receive local inhibition from GABAergic neurons. Finally, optogenetic activation of RMTg-GABA neurons in the LDT where Vglut2 is knocked down during REM sleep induced direct transition into NREM sleep.

The authors well examined the novel role of RMTg-GABA neurons which induce transition from REM sleep to Wake or NREM sleep. Particularly, the result that the activity of RMTg-GABA neurons gradually increased during REM sleep indicating reciprocal modulation of REM sleep was exciting. They should add some more information mentioned as follows:

Major comments:

- 1) In Figure 1D and 1E, the authors showed only REM sleep data. Wake and NREM sleep also should be involved in as Figure to confirm the effect of activation of RMTg-GABA neurons.
- 2) In Figure 2, 3, 4, 5 and 7, the authors performed optogenetic manipulation or electrophysiological recordings and sleep recordings. However, the timing of experiments such as light or dark phase is not described in the methods. The probability of each state can be differ depending on the circadian rhythm. Therefore, they should be careful about the timing and describe it in detail in the methods.
- 3) In Figure 2, 3, 5 and 7, the authors performed optogenetic manipulation. They described the timing of photostimulation in lines 932–936, but it is unclear whether the stimuli induced in vigilance state dependent manner or randomly, and what is the interval between the stimuli. Those conditions may affect the sleep architecture.
- 4) In Figure S4 experiment, isn't there a possibility that the increase of REM sleep amount within 4 hours (13:00–17:00) is just a rebound of REM sleep decreased during 2-h post-CNO injection period (9:00–11:00)? The results of the following optogenetic inhibition seem to be reliable, but those of the chemogenetic inhibition are hard to agree with the authors' current discussion in lines 600–605. The authors' idea would be described.
- 5) In Figure 5, the proportion of the first transition from REM sleep to non-REM and Wake would be involved to clearly show RMTg-GABA → LTD projection mainly induces transition to Wake, while RMTg-GABA → LH projection mainly does that to directly non-REM.

Minor comments:

- 1) Figure S1A “Merge”: The meaning of small white triangles should be described. If the triangles show coexistence of Foxp1 and GABA, there are 2 unnecessary ones in the bottom right.
- 2) Figure S1A and S1B: The length of scale bar should be described.
- 3) Figure S1C “Transforming” and “Rebuilding”: The method of “transforming” and “rebuilding” should be described in the methods.
- 4) Figure 1C and S4C: The baseline of the membrane potential and the firing rate is quite different. It seems that command current was injected in either or both experiments. The methods should be described in more detail. Additionally, statistical analysis such as firing rate before and after CNO application would be involved to clearly show the effectivity of chemogenetic manipulation.
- 5) Figures showing “sleep-wake state changes and probability of brain state transitions after photostimulation”, such as Figure 2E: Whether these Figures show data from all animals or a representative animal should be described.
- 6) Figures showing “cumulative probability”, such as Figure 2G: Calculation of the cumulative probability should be described in the methods since those Figures are confusing. For example, Figure 2E “ChR2” showed immediate termination of REM sleep by photostimulation. Therefore, I assumed that “cumulative probability” would immediately increase in the first tens of seconds in Figure 2G “REM termination” (like Weber et al. (2018) Nat commn. Figure 1f “REM→Wake” <https://doi.org/10.1038/s41467-017-02765-w>). However, the actual Figure showed gradual increase of “cumulative probability” through 120 sec.
- 7) Lines 191–192: “ChR2 group” and “the control group” are opposite.
- 8) Figure 4A “spike waveforms”: If the waveforms are from individual units, Figures of each waveform should be clearly separated.
- 9) Figure 4: Figures of in vivo electrophysiological recording sites would be involved.
- 10) Figure 5H “561 nm”: Before 561 nm stimulus period, is this really REM sleep? The value of delta band of EEG power spectrum seems to be higher compared to that of “473 nm” data below.
- 11) Figure 6N: The latency of the response seems too fast in some data which are less than 1 ms

compared to other data such as Figure 6D and 6I. Are these responses surely inhibited by picrotoxin? Wasn't there a possibility that ChR2 itself ectopically expressed in the cholinergic neurons?

12) Figure 7B: Did these slices express ChR2 or not? If ChR2 are expressed, the annotation "Glut" would be incorrect.

13) Line 659: "knockout" should be "knock down".

14) After manipulation of RMTg-GABA neurons to change the sleep architecture, were there any rebounds of each vigilance state?

Point-by-point responses to referees

Dear reviewers,

We appreciate the reviewers' thoughtful evaluations and insightful comments on our manuscript entitled "GABAergic neurons in the rostromedial tegmental nucleus are essential for rapid eye movement sleep suppression" (Manuscript Number: NCOMMS-21-43982A).

We have carefully addressed all the comments in this rebuttal letter and these constructive comments greatly improved our manuscript. Our replies are written in blue, and the line and page numbers of the corresponding revisions in the revised manuscript were provided. We hope that this rebuttal letter is satisfactory for reviewers.

Sincerely yours,

Su-Rong Yang

REVIEWER COMMENTS

Reviewer #1 (Remarks to the Author):

This manuscript found that RMTg GABAergic neurons promote REM sleep to wake transitions through disinhibition of glutamatergic LDT neurons. The experiments are rigorous, the manuscript is well written, very organized and uses many approaches to address the role of GABAergic RMTg neurons in REM sleep control. It is of great interest to the sleep field as well as the broader neuroscience community. The comments below aim to help clarify the important results and interpretations of the manuscript.

Major comments:

1. - Excitatory M3Dq experiments were conducted during the light period (inactive period). Given that exciting the RMTg previously was shown to promote sleep, please postulate if doing this experiment during the dark period (active period) have given a different outcome?

Reply:

Thank for the comments. We carried out the chemogenetic experiments during both inactive and active periods. Similar to the result that we observed during the inactive period, chemogenetic activation of RMTg GABAergic neurons during the dark period significantly promoted NREM sleep for about 4 h following clozapine-N-oxide (CNO) injection, but did not significantly decrease REM sleep, which may be due to less REM sleep in dark phase in control group under physiological condition.

Response Figure 1 Chemogenetic activation of RMTg GABAergic neurons during the active period promoted NREM sleep. a, b Sleep-wake quantities following administration of saline or clozapine-N-oxide, including the average hourly (a) and total sleep-wake amounts of REM sleep ($T_4 = 2.168$, $p = 0.0961$) during the 3-h post-injection period (21:00-24:00), and NREM sleep ($T_4 = 3.561$, $p = 0.0236$) and wakefulness ($T_4 = 3.478$, $p = 0.0254$) during the 4-h post-injection period (21:00-1:00) (b). * $p < 0.05$, ** $p < 0.01$. Statistics by two-way repeated ANOVA followed by paired t test in a and by paired t test in b. Data represent mean \pm SEM. $n = 5$.

2. -On page 7, lines 112-114, the efficacy of the chemogenetic techniques was established. However, there is no verification of their specificity in GABAergic neurons. Do the authors have histology with immunohistochemistry for GABA neurons to confirm that the stimulated neurons are GABAergic? i.e. Verifying the specificity of the vGAT-cre mouse line used.

Reply:

Thank for the suggestion. We observed the co-expression of mCherry+ neurons with GABA in VGAT-Cre mice microinjected with AAV-hM3Dq-mCherry. We created Fig. 1b showing the co-expression rates in 4 mice. Accordingly, we added the following text to results on lines 104-107, page 6 “Confocal images of double labeling (yellow) with mCherry (red) and GABA (green) immunofluorescence showed that 92% of the

hM3Dq/mCherry-positive neurons co-expressed GABA, validating the specificity of this targeting strategy for RMTg GABAergic neurons”.

Fig. 1b Representative photomicrographs of RMTg GABAergic neurons from a VGAT-Cre mouse microinjected with AAV vectors encoding hM3Dq. The mCherry (red) and green immunolabeling indicates hM3Dq- and GABA-expressing neurons of the RMTg, respectively, and the yellow image depicts merged neurons. Top: lower magnification of the images. The dashed circle area shows the location of the RMTg. Bottom: higher magnification of the dashed square region in the top corresponding images.

3. - Page 7, Line 117: the CNO dose used in mice should be listed somewhere at least once near the beginning of the results. It is mentioned for the patch clamp CNO dose used but seems to be missing for the behavioral experiment results *in vivo*.

Reply:

Sorry for missing the CNO dose for behavioral test. CNO was injected at a dose of 1 mg/kg in mice *in vivo*. We added it on line 114, page 6.

4. -On page 8, lines 120-126, the authors say that activation of RMTg caused fewer REM sleep periods, not shorter REM sleep periods (Figure 1). However, on page 16, lines 242-245, it is shown that inactivation of RMTg causes longer REM sleep periods, as opposed to more REM sleep periods (Figure S4). The authors then suggest that this second result indicates that RMTg GABAergic neurons inhibit REM sleep maintenance, but that is inconsistent with the first result. The second result is brought up in the Discussion (page 42, lines 596-597), but the first result's conflict with this needs to be addressed.

Reply:

Thank for the comments. Our results using AAV-hM3Dq suggested that the RMTg inhibited REM sleep initiation (Fig. 1), while the results using AAV-hM4Di suggested the role of the RMTg in suppressing REM sleep maintenance (Fig. S6). For optogenetics, activation of RMTg GABAergic neurons suppressed REM sleep maintenance and REM sleep initiation (Fig. 2). While optogenetic inhibition of RMTg GABAergic neurons prolonged REM sleep and facilitated REM sleep initiation (Fig.

3).

Different from chemogenetics, optogenetics has better temporal precision. Laser stimulation can be given in specific brain states. For example, in the present study, to observe the direct effects of RMTg GABAergic neurons on the initiation and maintenance of REM sleep, photostimulation was selectively given during NREM sleep and REM sleep, respectively.

We added the above information in the first part of the discussion, on lines 651-688, pages 42-43.

5. - Page 16, Line 242-245: Another possible interpretation of the inhibitory DREADDs data is that CNO led to increased wakefulness, but that the increased sleep drive (due to the experiment being carried out between 9:00-17:00/ light period) may have made even a mild 2 hour sleep deprivation period enough to lead to a minor sleep rebound?

Reply:

Thank you for the comment. We conducted an additional experiment to confirm that 2 h sleep deprivation (SD) cannot induce sleep rebound during the light phase. AAVs encoding hM4Di fused to mCherry were introduced into bilateral RMTg in VGAT-Cre mice. After 2-week recovery, electrodes for EEG recording were implanted. On the first day, saline was administered on 9:00 and EEG signals were recorded for 24 h used for baseline. On the second day, following saline administration on 9:00, sleep was deprived for 2 h by gentle handling. Similar to the hM4Di experiment, we observed that 2-h SD (9:00-11:00) significantly decreased REM and NREM sleep and increased wakefulness (Fig. S7a, b, left). However, within the 2 h after the end of SD (11:00-13:00), the amount of REM sleep was still significantly less than that of the control group (Fig. S7a). During the 4-h (13:00-17:00) after ending SD, there was no rebound of REM sleep in SD mice (Fig. S7b, right). In contrast, REM sleep rapidly reached the same level as the saline group from 11:00 to 13:00 in mice with chemogenetic inactivation of RMTg GABAergic neurons after CNO injection at 9:00, which induced a decrease of REM sleep from 9:00 to 11:00. Moreover, during 4-h from 13:00 to 17:00, REM sleep in mice with inactivation of RMTg GABAergic neurons significantly increased than saline group (Fig. S6).

Given that the biological effects of CNO could maintain for 6-10 h [1], during the period from 9:00-17:00 after CNO injection, the rapid and significant promotion of REM sleep following an increase in wakefulness for 2 h was due to inactivation of RMTg GABAergic neurons.

We added the information in results on lines 256-267, pages 16; and a paragraph in the discussion on lines 673-684, page 43.

Fig. S7 Sleep deprivation for 2 h did not cause any rebound of REM and NREM sleep in VGAT-Cre mice. **a** The hourly average amount of each stage [two-way repeated ANOVA followed by a *post hoc* Sidak test: $F_{1,8} = 1119.1$, $p < 0.01$ (REM); $F_{1,8} = 40.55$, $p < 0.01$ (non-REM); $F_{1,8} = 50.35$, $p < 0.01$ (wake) during 9:00–13:00] after saline administration at 9:00 and followed by 2-h sleep deprivation (SD). **b** Total amount of each stage during the 2-h SD period (9:00–11:00, left, REM sleep: $T_7 = 3.169$, $p = 0.0157$; NREM sleep: $T_7 = 3.361$, $p = 0.0121$; Wake: $T_7 = 3.531$, $p = 0.0096$) and 4-h period after saline injection (13:00–17:00, right). * $p < 0.05$, ** $p < 0.01$ vs. saline not SD group. Statistics by paired *t* test. Data represent mean \pm SEM. $n = 5$.

6.-One of the claims of the manuscript is that RMTg GABAergic neurons disinhibit LDT glutamatergic neurons (e.g., page 38, lines 537-538). However, RMTg GABAergic neurons may also synapse directly onto LDT glutamatergic neurons, as they do onto the other two types of LDT neurons, potentially exerting both disinhibitory and inhibitory effects on LDT glutamatergic neurons. Figure 6 shows the direct synaptic connections of RMTg neurons and ChAT/VGAT neurons, but does not show the IPSCs resulting from RMTg synapses onto glutamatergic neurons. It would be helpful for their conclusions if the authors included this data.

Reply:

Thanks for the suggestion. We conducted an additional experiment and found that RMTg GABAergic neurons also formed direct synapses with LDT glutamatergic neurons. The proportion of recorded LDT glutamatergic neurons that responded to the photostimulation of ChR2-expressing RMTg GABAergic terminals was 59%. The method in detail was added on lines 474-501, pages 31-32. These results were illustrated as Fig. 6k-o.

The present *in vitro* electrophysiological recording illustrated that RMTg GABAergic neurons, directly and indirectly, synapse onto LDT glutamatergic neurons, and therefore potentially exert both disinhibitory and inhibitory effects on LDT

glutamatergic neurons. When LDT glutamatergic neurons were knocked down, REM sleep was still rapidly terminated, with REM-NREM sleep conversions rather than transitions to wakefulness. Therefore, activation of RMTg GABAergic neurons may suppress REM sleep by direct inhibition of LDT cholinergic/GABAergic neurons instead of glutamatergic neurons and mediates REM sleep-arousal transitions by disinhibition of LDT glutamatergic neurons. The above information was provided in the discussion on lines 723-738, pages 45-46.

Fig. 6k–o RMTg GABAergic neurons directly inhibited LDT glutamatergic neurons. **k** Schematic diagram of the experimental protocol. In VGAT-Cre mice, Cre-dependent AAVs encoding ChR2 were injected into the RMTg, whereas AAVs encoding the promoter of cholinergic neurons (ChAT-eGFP) and DIO-eGFP were injected into the LDT of VGAT-Cre mice. **l** Representative images showing a recorded biocytin-filled neuron (violet) that is a glutamatergic neuron with no eGFP expression. Scale bar: 50 μ m. **m** Typical traces of inhibitory postsynaptic currents (IPSCs) from an LDT glutamatergic neuron evoked by blue light stimulation (black line) of ChR2-expressing GABAergic terminals in the RMTg. The evoked currents were completely abolished by application of picrotoxin (PTX) (red line). **n** Latency of light-evoked IPSCs in LDT glutamatergic neurons that responded to laser stimulation of ChR2-expressing GABAergic terminals in the RMTg. Data represent mean \pm SEM. **o** Proportion of recorded LDT glutamatergic neurons that responded to laser stimulation of ChR2-expressing GABAergic terminals in the RMTg. $N = 17$ cells from 3 mice.

7. -Please elaborate in the discussion the apparent paradox of the RMTg GABAergic neurons having high firing rates during wake but when activated with optogenetics promote NREM sleep.

Reply:

Thank you for the question. For wake-promotion neurons such as dopaminergic neurons in the ventral tegmental area (VTA) and GABAergic neurons in the ventral pallidum (VP), the population activity increased to a higher level before the transition from NREM sleep to wakefulness. That means higher neuronal activity at the end of the NREM sleep state is needed to generate arousal [2,3].

Differently, there is an increase in firing rates 30 s before the transition from wakefulness to NREM sleep transition. Furthermore, RMTg GABAergic neurons have distinct discharge patterns like a kind of phasic firing, in which spiking frequency peaks corresponded to EEG delta oscillation (Fig. S8), which may also contribute to NREM sleep-promotion. In the present study, we found that the RMTg GABAergic neurons have high firing rates during wake but when activated with optogenetics promote NREM sleep, which is in an agreement with NREM sleep-promotion of vIPAG and

VTA GABAergic neurons that have similar firing profiles of higher neuroactivity when at awake state [4,5].

During wakefulness, dopaminergic neurons in the VTA and GABAergic neurons in VP are at high levels of neuronal activities, we think the high firing rates of RMTg GABAergic neurons during wake were to protect the brain from overactivation. While during NREM sleep, the wake-promoting neurons in the VTA and the VP are at low activity, and thus, RMTg neurons may not need a higher level of activity. However, to drive NREM sleep from wakefulness, these wake-active neurons must be counteracted by optogenetic activation of RMTg GABAergic neurons. The exact relationship of neuronal firing levels between NREM sleep-promoting neurons in the RMTg and wake-promoting neurons needs further investigation.

We have added a paragraph about spike firing patterns during brain state transitions and related analysis in sleep-wake regulation in the discussion section on lines 707-720, pages 44-45.

Fig. S8 RMTg GABAergic units firing characteristics across distinct sleep-wake

states. a Firing rate modulation of 12 identified units from 3 mice (z-scored). W: wake, R: REM sleep, N: NREM sleep ($p < 0.05$, Wilcoxon rank-sum test). **b** Firing rates of significant NREM-active (blue) and NREM-inactive (red) neurons during different brain states. Each line shows the firing rates of one unit; bar, average firing rates of the units. **c–e** Autocorrelograms for firing patterns of RMTg GABAergic units in each brain state. Each row represents one trial of REM sleep (c), NREM sleep (d), and wakefulness (e) (top). Average of autocorrelation of each brain state (bottom). Shading, 95% confidence intervals. **f** Representative firing rates of a RMTg GABAergic neuron together with EEG spectrogram and EMG amplitude. Green lines indicate time point of high delta oscillation (0.65–4 Hz). **g** Spike-EEG spectrogram coherence of each identified unit. **h** Average coherence in g. Shading, 95% confidence intervals.

8. -Pages 58-59, lines 932-942. Although the authors specify that wakefulness, REM, and non-REM sleep were scored offline using SleepSign 3.0, they do not describe how this is evaluated online for the optogenetic stimulation experiments. Furthermore, they specify that stimulation during REM periods happens ≥ 16 s after the start of REM (or ≥ 60 s for NREM), but not how the timing of the optogenetic stimulation was determined beyond that. The figures show that sometimes the stimulation occurred long after 16 s, but other times it was shorter. Please describe the criteria for online sleep scoring and stimulation timing.

Reply:

The photostimulation experiments were mainly performed during the light period. The laser was randomly delivered in different brain states, namely either REM sleep, non-REM (NREM) sleep, and quiet or active wakefulness, with an interval of at least 10 min given that REM sleep occurs every 10 to 20 min in mice [6]. EEG signals were scored offline using SleepSign 3.0 to determine different vigilance states. All trials that light stimulation was initiated when mice naturally went into REM sleep for at least 16 s or NREM sleep/wakefulness for at least 60 s were used for analysis [7].

We added a paragraph about the method in detail for the optogenetic stimulation experiments on lines 939-949, pages 56-57.

Minor comments:

-Page 1, line 16. Please change disorder to disruption or disturbances. “REM sleep disorder” is not a defined diagnosis and sounds a lot like “REM sleep behavior disorder” which I don’t think you are talking about.

Reply:

Thank you for the suggestion. We corrected.

- Page 13, Line 185: These optogenetic experiments used 120 seconds of stimulation to see changes in sleep states. The authors showed previous data above testing different frequencies of stimulation, did they also test other durations of stimulation time?

Reply:

Thank you for the suggestion. We tested different durations of stimulation time including 30 s, 60 s, 120 s, and 300 s. There are significant differences between the Chr2 and mCherry groups regardless of stimulation time regarding REM sleep transition latency and NREM sleep duration ratio when the laser was illuminated during NREM sleep and wakefulness, respectively. In addition, the NREM sleep duration ratio increased from the 30 s group to the 120 s group, but it decreased from the 120 s group to the 300 s group although there are no significant differences (Fig. S4). Thus, a duration of 120 s was chosen for further optogenetic stimulation study.

We added the above information on lines 229-233, page 15.

Fig. S4 REM sleep transition latency and NREM sleep duration ratio with different durations of laser stimulations in VGAT-Cre mice. a REM sleep transition latency with laser stimulation during REM sleep. **b** NREM sleep duration ratio, calculated as NREM sleep duration during the time of laser stimulation divided by time of laser stimulation during wakefulness. Mann–Whitney test between the Chr2 group and mCherry group. Wilcoxon signed rank test within the Chr2 group. * $p < 0.05$, ** $p < 0.01$. Data represent mean \pm SEM. Chr2: $n = 5$, mCherry: $n = 4$.

-On page 13, line 191, I think “Chr2” and “control” were accidentally swapped.

Reply:

Sorry for the mistake. We corrected.

-On page 13, line 199, should “ms” be “s”?

Reply:

Sorry for the mistake. We corrected.

- Page 19, Line 282: why was only 60 seconds laser stimulation used in inhibitory optogenetic experiments? Usually it is much harder to push a system using inhibition compared to excitation (120s used for Chr2 experiments).

Reply:

The protocol for activating ChR2-expressing neurons was to use 120 s photostimulation at 30 Hz, which means 10 ms per pulse, followed by a 23 ms interval. Unlike the excitation modality, the inhibition protocol uses 60 s of constant light. The inhibition protocol for Arch-expressing neurons was chosen based on the literature: ①Zhang Z, cell, 2019; ②Chen KS, neuron, 2018. These two references have been added in the method section on lines 938-939, page 56.

-Likewise, on page 22, line 326-327, should “10 ms” be “10 s”? A 30 Hz stimulation applied every 10 ms doesn’t make much sense.

Reply:

Sorry for the confusion. We gave light stimulation for 10 s at 30 Hz (10 ms per pulse). This information has been added on lines 340-341, page 21.

-Page 24, line 356: did tracing with unilateral microinjections label the contralateral RMTg or ipsilateral side?

Reply:

Thank you for the suggestion. We found that unilateral microinjections of anterograde tracer eGFP into the RMTg labeled: ①The RMTg ipsilateral side of the SNc, LH and CL, however, labeled the RMTg contralateral side of the GiV; ②Both RMTg ipsilateral and contralateral side of the VTA and LDT, but with more projections to the ipsilateral side; ③Almost the same projections to the RMTg ipsilateral or contralateral sides of the MnR, RMg, PnC, and the DR.

In the optogenetic experiments to observe the neural circuits of REM sleep suppression induced by activation of RMTg GABAergic neurons, we injected AAV-DIO-ChR2-mCherry into bilateral sides of the RMTg and implanted optical fibers in bilateral sides of the LDT or LH, thus excluding the influence of different projection patterns at different targets of RMTg GABAergic neurons.

-Page 24, line 358: define abbreviations

Reply:

Thank you for the suggestion. We defined.

-It is unclear based on page 26, line 382, if the authors are stimulating GABAergic cell bodies or just nerve endings. The schematic in Figs. 5A and 5G make it look like the latter, so please correct line 382 if needs be.

Reply:

Sorry for the mistake, it has been corrected on lines 400-401, page 26. We stimulated only RMTg GABAergic nerve endings.

-Page 26, line 384: what frequency and duration was laser stimulation?

Reply:

The frequency of laser stimulation was 30 Hz with 10 ms per pulse, duration of laser stimulation was 120 s. This information has been added on lines 403, page 26.

-Page 36, line 507: with respect to the optogenetic terminal patch clamp experiments carried out in the LH, it would be useful to mention if GABAergic neurons make up a large proportion of LH neurons or is this a subpopulation that the authors were trying to look at?

Reply:

Thank for the comment. In the LH, there are mainly four types of neurons, including GABAergic, glutamatergic, melanin-concentrating hormone (MCH), and orexin neurons [8]. In the manuscript, we have found that RMTg GABAergic neurons formed synapses with LH GABAergic neurons, which comprise a big proportion of LH neurons. Now we conducted additional experiments to find whether there are functional connections from RMTg GABAergic neurons to the other three types of neurons in the LH.

We injected AAV-DIO-ChR2 into the RMTg and a mixture of viruses including DIO-eGFP, AAV-orexin-eGFP, and AAV-MCH-eGFP was injected into the LH in VGAT-Cre mice. In the LH, the neurons expressing eGFP were GABAergic, orexin, and MCH neurons. Neurons that did not express eGFP were assumed to be glutamatergic neurons. In vitro recordings showed that there are 48% direct synaptic inhibitory connections from the RMTg GABAergic terminals to the LH glutamatergic neurons. Similarly, there are 32% synaptic connections from the RMTg GABAergic terminals to the LH orexin neurons. However, the light-evoked IPSCs in the LH orexin neurons were produced in longer than 5 ms, indicating an indirect synaptic connection. Surprisingly, there were no synaptic connections from RMTg GABAergic terminals to the LH MCH neurons. These results were shown in Fig. S11f–s.

Taken together, these results suggested that activation of RMTg GABAergic nerve endings in the LH may promote REM-NREM sleep transitions through direct inhibition of GABAergic and/or glutamatergic and/or indirect inhibition of LH orexin neurons in the LH.

The related results (lines 625-634, pages 40-41) and discussions (lines 739-765, page 46-47) were added.

Fig. S11 Direct inhibitory connections between RMTg GABAergic neurons and LH GABAergic and glutamatergic neurons. **a, f, k, p** Schematic diagrams of the experimental protocols. Cre-dependent adeno-associated viruses (AAVs) encoding Chr2 were introduced into the RMTg, whereas Cre-dependent AAVs encoding eGFP (**a**); three kinds of AAVs encoding eGFP, promoter of orexin neurons (ORX-eGFP), and promoter of melanin-concentrating hormone (MCH) neurons (MCH-eGFP) (**f**); AAVs encoding ORX-eGFP (**k**), and MCH-eGFP (**p**) were injected into the lateral hypothalamus (LH) of VGAT-Cre mice. **b, g, l, q** Representative images showing a recorded biocytin-filled neuron (violet) that is a GABAergic neuron (eGFP: green) (**b**), a glutamatergic neuron with no eGFP expression (**g**), an orexin neuron (ORX-eGFP) (**l**), and MCH neurons (MCH-eGFP) (**q**). Infected RMTg GABAergic terminals in the LH (mCherry: red) were optogenetically activated. Scale bar: 50 μ m. **c, h, m, r** Typical traces of postsynaptic currents from an LH GABAergic (**c**), glutamatergic (**h**), orexin (**m**), and MCH (**r**) neuron evoked by blue light stimulation (black line). The evoked currents were completely abolished by application of picrotoxin (PTX) (**c, h, m, r**, red line). **d, i, n** Latency of light-evoked postsynaptic currents in LH GABAergic (**d**), glutamatergic (**i**), and orexin (**n**) neurons. Data represent mean \pm SEM. **e, j, o, s** Proportion of recorded LH GABAergic (**e**), glutamatergic (**j**), and orexin (**o**) neurons that responded to laser stimulation of ChR2-expressed RMTg GABAergic terminals, while MCH neurons did not respond to blue light stimulation (**r, s**). $N = 21$ cells (**e**), $N = 23$ cells (**j**), and $N = 19$ cells (**o**) from 3 mice, and $N = 24$ cells (**s**) from 4 mice.

-Page 55, line 866: It would improve the manuscripts scope if both sexes were used, given sex-related differences in sleep (e.g., see Dib et al., 2021, *Neurobiol Sleep Circadian Rhythms*). Otherwise, the Results section needs to be specific when presenting the *in vivo* results that they are from male mice only.

Reply:

Thank you for the suggestion. In our manuscript, all *in vivo* results observing sleep–wake behavior by EEG recording were from male mice only. Dib, et al (2021) review studies about sex differences in sleep and circadian phenotypes in mice and rats. They think gonadal hormones, sex chromosomes, and brain organization differences in the expression of gonadal hormone receptors in sleep and circadian-related brain areas could all contribute to sex differences [9]. Therefore, in future research, female rodents are needed to be used to investigate whether RMTg GABAergic neurons exert suppressing REM sleep as in male mice.

In the method section, we pointed out that in all *in vivo* experiments observing sleep–wake behavior, we used male mice on lines 856, page 52.

-Please add a line explaining why you didn't continue to RMTg neurons projecting to midbrain dopamine regions like the VTA and SN

Reply:

Thank for the suggestion. Increasing evidence suggests that dopaminergic neurons are responsible for wakefulness [2,7,10,11]. Whether dopaminergic neurons are involved in REM sleep regulation is poorly understood. Eban-Rothschild, et al (2016) found that chemogenetic inactivation of VTA dopaminergic neurons increased the number of REM sleep episodes [2]. Differently, no change in REM sleep was observed when VTA/SNc dopamine neurons were inactivated in our previous study. In addition, our previous study revealed inhibitory connections from the RMTg GABAergic neurons to midbrain dopaminergic neurons [10]. In the present study, we found that activation of RMTg GABAergic neurons immediately terminated REM sleep and starts brief arousal. Based on the findings from Dr. Eban-Rothschild and us, inhibition of midbrain dopaminergic neurons by RMTg GABAergic neurons may increase REM sleep or produce no change. Very recently, Sakurai et al (2022) found that a transient increase in dopamine levels in the basolateral amygdala from the dorsal region of the VTA during NREM sleep initiates REM sleep, while from the ventral part of the VTA projecting to the nucleus accumbens does not promote REM sleep [12]. Since the RMTg forms inhibitory synapses with VTA dopaminergic neurons that preferentially target the nucleus accumbens (16), the RMTg^{GABA}-VTA circuit may not be the primary pathway mediating the suppression of REM sleep induced by RMTg GABAergic neurons.

Therefore, we did not investigate the function of RMTg^{GABA}-VTA/SNc pathways in REM sleep termination. Instead, the role of RMTg^{GABA}-LDT/LH pathway in REM sleep is further studied based on virus anterograde tracing results combined with the REM sleep-promoting functions of cholinergic neurons in LDT [13], and MCH [14],

orexin [15], and GABAergic [16] neurons in LH.

We added the information “combined with the wake-regulatory role of dopaminergic neurons revealed by growing evidence” on lines 392-393, page 25 in the result section.

-Was there blinding during any of the analyses or experiments?

Reply:

Yes, the investigator who checked the EEG signals to determine the brain states was blind to the group information. This sentence has been added in the method section line 930-931, page 56.

Figures:

-In Figure 1B (and other figures), please show more than one slice so readers can judge the extent of the anterior-posterior expression, maybe in supplementary figures.

Reply:

Thanks for the suggestion. We illustrated the anterior-posterior virus expression in the RMTg from 4 brain levels for each mouse in all behavioral studies. Fig. S2 showed that the virus expression was mostly restricted in the RMTg. In addition, the virus expression in the LDT when observing possible roles of LDT glutamatergic neurons in the REM-wake transitions induced by activation of RMTg GABAergic neurons was also illustrated at 3 brain levels, which showed that the virus was mostly restricted in the LDT.

Fig. S2 AAVs-microinjected area of each tested mouse in different behavioral experiments. **a-f** In the experiments observing sleep-wake states by manipulating RMTg GABAergic neurons (**a-d**) or the circuits of RMTg^{GABA}-LDT (**e, g**) and RMTg^{GABA}-LH (**f**), the coronal sections at four brain levels from bregma -3.80 mm to bregma -4.48 mm show the superimposed area of the RMTg GABAergic neurons expressing AAVs encoding hM3Dq (**a**), hM4Di (**b**), Chr2 (**c, e, f, g top**), and Arch (**d**)

and the superimposed area of the LDT neurons expressing shVglut2 at three brain levels from bregma -4.96 mm to bregma -5.20 mm (g, bottom) in each mouse.

-On Figure 1H, please include the units on the x-axis. Multiple t-tests were used for comparing groups at different time points but this seems like a situation for a two-way ANOVA? It's unclear how many mice were used for patch clamp experiments to confirm CNO activation of dreadd receptors vs. behavioral M3Dq experiments. Please add the n for these experiments.

Reply:

Thank you for the suggestions. Fig. 1H is now changed as Fig. 1i.

- (1) Fig. 1i, the unit on the x-axis "Duration (s)" was added.
- (2) Fig. 1i, the data has been analyzed using repeated two-way ANOVA followed by Sidak's multiple comparison test. This statistical result has been rewritten on line 124, page 7. The legend has been correspondingly corrected.
- (3) Patch-clamp experiments to confirm CNO activation of DREADD receptors, 7 cells from 3 mice, on line 148, page 9.
- (4) Behavioral hM3Dq experiments, $n = 9$ mice, on line 136, page 9.
- (5) The statistical analysis for Fig. 1i was corrected on lines 1071-1074, page 63.

-In Figure 2E, there appears to be some amount of rebound REM sleep happening after REM sleep is suppressed, relative to the same time period during the control conditions. Analysis of this effect could be interesting for determining the physiological function of these neurons.

Reply:

Thank for the suggestion. We analyzed the sleep/wake state transitions after the stop of photostimulation of the RMTg GABAergic neurons. In Fig. 2E, from about 90 s after the end of 120-s laser illumination, more REM sleep episodes occurred than that in the control group. These results also indicated the activity of RMTg GABAergic neurons is necessary for terminating REM sleep physiologically.

The result of the rebound analysis was shown in the following figure.

Response Figure 2 REM sleep rebound after the end of optogenetic activation of RMTg GABAergic neurons. During 300 s after the end of 120-s laser stimulation of RMTg GABAergic neurons, the probability of REM sleep after REM sleep inhibition in Fig. 2E. Black, control group. Magenta, REM sleep in ChR2 group; Red, values with

significant differences (Two-way ANOVA followed by a post hoc Sidak test). Shading, SEM.

-Figure 3C, this panel is slightly confusing visually because both example traces list -60mV at the start of the trace but one actually represents the membrane potential measured and one the changes in current. Maybe the labeling can be adjusted slightly.

Reply:

Sorry for the mistake. The bottom showed the changes in current when the target cell was held on the voltage of -60 mV. The labeling in Fig. 3c was corrected.

-Figure 4B: Please adjust the spectrogram color bar so you can see the oscillation dynamics more clearly

Reply:

Thank you for the suggestion. We adjusted the spectrogram color bar in Fig. 4B (now changed Fig. 4c).

Fig. 4c Firing rates of an example RMTg GABAergic neuron (yellow) together with the EEG spectrogram, EMG amplitude, and color-marked brain state.

-Figure 4 C-D: color scheme makes it hard to see the confidence intervals. Maybe darken these lines or use shading if possible?

Reply:

Thank you for the suggestion. We shaded the confidence intervals in Figure 4 C-D (now changed Fig. 4d, e).

Fig. 4d Mean firing rates of all identified GABAergic neurons at sleep–wake state transitions. **e** Firing rate changes of identified RMTg GABAergic neurons from NREM to REM sleep transition (left) and linear regression of firing frequency of all identified units during the initial 30 s of REM sleep (right). Dashed line: confidence bounds ($R^2 = 0.819, p < 0.01$). Shading indicates 95% confidence intervals in **d** and **e**. $n = 12$ units from 3 mice.

-Figure S5: it would be useful to give estimates of A-P location in the brain for slices shown

Reply:

The A-P location of the brain slices in Figure S5 (now changed Fig. S9) was added.

-Figure S6, Page 32, Line 442: “Colorful stairs... mCherry group...” do you mean control laser (yellow) vs blue light instead of ChR2 vs mCherry control group?

Reply:

Sorry for the confusion. In Fig. S6 (now changed Fig. S10), the mice were delivered AAV-ChR2-mCherry in the RMTg, then a blue laser with 473 nm was used to activate RMTg GABAergic terminals within the LDT/LH and a yellow laser with 561 nm was used as control.

-Figure 6: this figure is confusing because initially the panels A, F, K look identical but closer examination shows that A,F experiments are exciting RMTg terminals in the LDT while panel K is using the same viruses to eliminate GABAergic and cholinergic neurons in the LDT and instead record from glutamatergic LDT neurons? The Figure 6 legend text for panels K-O might need another sentence to clarify this part of the figure and distinguish it from the panels above it.

Also, Figure 6 and corresponding section 6 in the Results (page 32) might be clearer if they use “GABAergic RMTg,” “cholinergic LDT” and “GABAergic LDT” neuron labels were clarified in the text and figure legend.

Reply:

Sorry for the confusion. Fig. 6 has been reorganized as follows and the legend and related result (lines 474-501, pages 31-32) has been rewritten.

Fig. 6 RMTg GABAergic neurons directly inhibited LDT cholinergic, GABAergic, and glutamatergic neurons and indirectly disinhibited LDT glutamatergic neurons. a–o RMTg GABAergic terminals formed inhibitory connections with LDT cholinergic, GABAergic, and glutamatergic neurons. **p–t** Within the LDT, GABAergic terminals formed inhibitory connections with glutamatergic neurons.

-Figure 7 E-G: please clarify the labeling in the figure and legend text

Reply:

Sorry. We rewrote the labeling and legend text as follows.

Fig.7e Probability of first transitions from REM sleep to wakefulness by ChR2-mediated activation of RMTg GABAergic neurons compared with mCherry group. ChR2: $n = 5$, mCherry: $n = 4$. **f, g** Comparison between the probability of first transitions from REM sleep to wakefulness (**f**) and REM to NREM sleep (**g**) by activation of the RMTg^{GABA}-LH ($n = 5$), RMTg^{GABA}-LDT ($n = 5$), and RMTg^{GABA}-LDT with shVglut2 ($n = 5$) and shCtrl ($n = 4$) microinjection in the LDT, respectively. All the VGAT-Cre mice were delivered AAVs encoding ChR2 into the bilateral RMTg, with optic fibers in the LDT or LH. 561 nm, 561 nm yellow laser stimulation was used as the control group. 473 nm, 473 nm blue laser stimulation was used as the experimental group. * $p < 0.05$, Wilcoxon signed rank test for the experimental group with the 561nm control group; # $p < 0.05$, Mann–Whitney U test for the shVglut2 and shCtrl group. Data represent mean \pm SEM.

Reviewer #2 (Remarks to the Author):

The paper by Zhao et al. investigates whether and how GABAergic neurons in the rostromedial tegmental nucleus (RMTg) are involved in the suppression of REM sleep. At first, the authors labeled the Foxp1 immunoreactive neurons and found that Foxp 1 was expressed at higher levels and strongly colocalized with GABA in the RMTg. Consequently, the authors used viral approaches to target and manipulate the RMTg GABAergic neurons. The authors showed that optogenetic activation of the RMTg GABAergic neurons immediately converted REM sleep to arousal and then initiated NREM sleep. Likewise, chemogenetic activation of the RMTg GABAergic neurons decreased the REM sleep episodes. In contrast, optogenetic inactivation completely transitioned NREM to REM sleep and prolonged REM sleep duration. The REM sleep control was interpreted to a disinhibition effect on the LDT glutamatergic neurons. The involvement of RMTg GABAergic neurons in REM was also demonstrated by their firing properties during the Sleep–wakefulness transition.

The authors conducted many experiments to address these questions, utilizing state of the art genetic circuit dissection techniques and challenging experimental approaches. Considering their strong influence on brain states, the data shown are interesting and suggest a potential role for the RMTg GABAergic neurons in controlling REM sleep. However, I'm not completely convinced of main conclusions drawn from the current results because of the following concerns:

Major concerns:

(1) The RMTg GABAergic neurons receive major input from the lateral habenula, which conveys negative reward and motivation related information. Moreover, it has been demonstrated that the RMTg GABAergic neurons project to ventral tegmental area, dorsal raphe nucleus and locus coeruleus, regulating diverse functions such as aversive and despair behavior, addiction and pain. Why the RMTg GABAergic neurons are required to be activated during sleep and subserve for the suppression of REM sleep? How the RMTg GABAergic neurons are activated during sleep? More contexts should

be provided for supporting and interpreting their claims.

Reply:

Thanks for this fundamental and important question. Sleep is regulated by homeostasis and circadian factors. In addition, sleep is influenced by a variety of emotional and physiological factors, such as stress, pain, feeding, and body temperature [6]. It has been reported that acute social defeat stress activates VTA GABAergic neurons, which sense stress and drive sleep and consequently restore mental and body functions by sleeping to alleviate anxiety disorders [17,18]. REM sleep is well known as a brain state supporting negative memory consolidation [19]. On the other hand, stress-induced anxiety [20] and depression [21] are also associated with excessive activation of RMTg GABAergic neurons. Therefore, promoting NREM sleep while suppressing REM sleep is crucial for alleviating stress responses and protecting our brain from psychiatric disorders. We used RV retrograde tracing techniques and found that RMTg GABAergic neurons receive wide projections in the brain, including dense projections from the lateral hypothalamus (LH), lateral habenula, and zona incerta. Thus, RMTg GABAergic neurons may exert roles of sleep by the inputs from the above brain regions and then integrating the diverse information of aversive, despair behavior, and pain. However, what's kind of emotions or physiological factors such as feeding in which the LH is implicated [22] activate RMTg GABAergic neurons that trigger or maintain NREM sleep or terminate REM sleep need more investigation.

Anatomically, RMTg GABAergic neurons receive both excitatory and inhibitory afferents from many brain regions related to sleep–wake regulation. RMTg GABAergic neurons may exert sleep by being activated by sleep-related neurons or inactivated by wake-related neurons. For example, the LH provides the strongest dense presynaptic connections with RMTg GABAergic neurons [23], thus, we try to investigate this projection in the roles of the RMTg in sleep regulation. [REDACTED].

Moreover, some brain regions such as the PAG, DR, and LC that contain a large number of REM-off neurons send dense inputs to RMTg GABAergic neurons [23-25]. RMTg GABAergic neurons perhaps exert REM sleep suppression when receiving information from these REM-off input nuclei.

We added a paragraph about possible mechanisms for controlling RMTg GABAergic neurons to regulate sleep in the discussion on lines 834-843, page 51.

[REDACTED]

[REDACTED]

(2) Optogenetic and chemogenetic manipulation of the RMTg GABAergic neurons are used as a major argument for their causal role in REM sleep regulation. However, it is not mentioned how the authors controlled the location specificity of their virus injections. Because the RMTg is a rather elongated and irregular structure (see Fig. S1B), it is hard to know for sure that all RMTg and only RMTg are injected with virus. Indeed, as shown in Fig. 1B, one of its immediate neighbors, the VTA is obviously transfected by the virus, which thus weakens their claims on the role of RMTg GABAergic neurons in REM sleep control. Data strictly viral expressing in the RMTg for each tested mouse would be helpful to strengthen their claims.

Reply:

Thanks for the suggestion. Indeed, up to now, there is no specific RMTg location on the mouse brain atlas. It has been found that the transcription factor *Foxp1* is highly expressed in the RMTg compared with their surroundings [26,27]. Thus, to determine the location of the RMTg in mice, we first did the immunohistochemistry of *Foxp1*. The relative location of the RMTg was produced on the mouse atlas according to the range of *Foxp1* expression, which spreads from bregma at -3.40 mm to -4.84 mm (**Fig. S1**). The location of the RMTg defined by *Foxp1* revealed that the caudal VTA, a part of the anterior tegmental nucleus (ATg), the pontine reticular nucleus, and the superior central nucleus raphe were included in the RMTg region, which is consistent with the previous study [28].

We illustrated the anterior-posterior virus expression in the RMTg from 4 brain levels for each tested mouse in all behavioral experiments (**Fig. S2 on page 15 of this rebuttal letter**). This supplementary figure showed that the virus expression was mostly restricted in the RMTg. We found there are virus expressions in the caudal VTA on bregma -3.80 mm, which is factually included in the newly defined region of the RMTg. In addition, a small amount of virus expression was found in the pontine reticular nucleus outside of the RMTg, but previous studies showed that GABAergic

neurons in the pontine reticular nucleus are related to REM sleep promotion [29-33], which is opposite to the effect of RMTg GABAergic neurons.

Overall, the injected virus was mostly restricted in the newly defined region of the RMTg, which corresponded to prior mouse studies of the RMTg [28,34-36].

(3) Firing rates of the RMTg GABAergic neurons at brain state transitions are used as another major argument for their role in REM sleep control. Based on their claims, the RMTg GABAergic neurons might function as REM-off units. However, the activity of RMTg GABAergic neurons significantly increased during the transition from NREM to REM sleep (Fig 4D). How to explain these contradictory results? Likewise, how to explain the decreased firing of RMTg GABAergic neurons during the wake-NREM transition (Fig. 4C)? As shown in Fig. S3A, optogenetic activation of RMTg GABAergic neurons resulted in increased NREM sleep.

Reply:

Thank for the question. As shown in Fig. 4D (now changed Fig. 4e), the neuronal activity of RMTg GABAergic neurons was lower at the onset of REM sleep and gradually increased to reach relatively high levels at the termination of REM sleep. During NREM and REM sleep, activation of RMTg GABAergic neurons with blue laser inhibited REM sleep generation and immediately terminated REM sleep, respectively. These results clearly indicated that these neurons functions as REM-off neurons from sleep–wake behavior regulation. The temporal change of neuronal activity across brain states and the roles in gating REM sleep of RMTg GABA neurons are consistent with the periaqueductal GABAergic neurons [4]. This phenomenon can be interpreted as homeostatic regulation of REM sleep. REM sleep propensity accumulates in NREM sleep and discharges during REM sleep [37]. From the mutual inhibition model for REM sleep regulation, with the release of REM sleep pressure, the inhibition of the REM-off neurons became progressively decreased, as a result, the firing of REM-off neurons reaches high levels at the end of REM sleep and induced the termination of REM sleep [38-40]. Like adenosine, although it accumulates and gradually increases in the awake state, it functions as a sleep-promotion substance [41].

The second concern is the same with Q7 from the first reviewer. **Please refer to the reply and Fig. S8 on page 6-8.** We added a paragraph about spike firing pattern during brain state transitions and related analysis in sleep–wake regulation in the results (lines 374-381, pages 24-25) and discussions (lines 707-720, page 45).

(4) Did all RMTg GABAergic neurons showed similar firing pattern during brain state transitions. The authors should make careful analysis for each recorded RMTg GABAergic neurons in Fig.4, and compared their firing activity across distinct states. This analysis is necessary because the RMTg GABAergic neurons project to diverse downstream targets and play various roles as recently reported.

Reply:

Thank for the suggestion. We analyzed firing rates and firing patterns of RMTg

GABAergic neurons across distinct Sleep–wake states. Fig. S8 (on page 7 of this rebuttal letter) showed that there are two clusters of RMTg GABAergic neurons firing at high rates (36 Hz) and low rates (3 Hz) in NREM sleep, respectively. We further analyzed the firing patterns of these identified neurons [42,43]. Autocorrelation across different Sleep–wake states illustrated that the discharge of RMTg neurons peaks at 20 ms and 240 ms during NREM sleep, respectively, which corresponded to 50 Hz and 4 Hz, respectively. This firing profile showed a distinct firing pattern during NREM sleep compared with those during REM sleep or wakefulness. This firing pattern revealed a kind of phasic firing. Moreover, coherence analysis revealed that the spiking frequency peaks corresponded to EEG delta oscillation at 0.65 and 4 Hz.

Different neuronal firing patterns play varied roles. For example, it is found that a depression-like state depends on a bursting mode of firing in the lateral habenula neurons [44]. Here, coherence analysis showed that this kind of burst-like firing pattern is related to synchronization since the spiking frequency corresponds to EEG delta oscillation. Considering that RMTg neurons project to diverse downstream targets and implicated various functions, thus, analysis of firing patterns under different situations may help elucidate the mechanisms of related behaviors.

The spike firing pattern during brain state transitions and related analysis in sleep–wake regulation was added in the results on lines 374-381, pages 24-25, and in the discussion on lines 707-720, page 45.

(5) The downstream LH neurons for the RMTg GABAergic neurons should also be identified which is critical for thorough understanding their hypothesis listed in Fig. 8.

Reply:

Thank you for the suggestion. This question is related to the question from the 1st reviewer, please refer to the reply on pages 11-12 and Fig. S11.

It is known that the LH is composed of largely heterogeneous cell types and circuits. Although LH neurons are mainly identified by the expression of specific markers, such as orexin and MCH, single markers cannot account for LH cell type diversity. The neuronal cell types can be further classified according to some specific markers, such as Lhx9, dynorphin, lepRb, and neurotensin. Some subpopulations overlap. These neuropeptidergic subpopulations are thought to also express GABA or glutamate. Different populations project and interact with each other, for example, a subtype of LH GABAergic neurons project to and inhibit orexin neurons, but the exact synaptic microcircuit between the LH neurons has not been fully understood [22,45-47]. Therefore, it is a big challenge to clarify the specific neuron types that mediate the REM-NREM transitions induced by activation of RMTg^{GABA}-LH pathway.

The related results (lines 625-634, page 40-41), and discussion (lines 739-765, page 46-47) were added to the manuscript.

Minor points:

(1) Given the strong influence of optogenetic inactivation, it is hard to understand why chemogenetic inactivation of the RMTg GABAergic neurons only plays subtle role in

initialize or stabilize REM sleep as shown in Fig S4D, E. More explanations should be made for these results.

Reply:

Thank you for the question. In optogenetic experiments, we give yellow photostimulation of RMTg GABAergic neurons at specific brain states of NREM or REM sleep. Possibly due to strong effects on termination of REM sleep under physiology situations, so we found inactivation of these neurons produced immediate NREM-REM transitions and maintained REM sleep, respectively.

In contrast, with inactivation by chemogenetics, it is hard to observe transitions between different brain states due to a lack of temporal precision compared with optogenetics. To evaluate the effects of REM sleep-promotion by inactivation of RMTg GABAergic neurons, we performed an additional experiment to deprive mice of sleep. We found 2-h sleep deprivation (SD) by gentle handling from 9:00 to 11:00 induced about 4 hours of REM sleep decrease within 9:00-13:00. During the following 4 h from 13:00-17:00 after the end of SD, the deprived mice showed no augmentation of REM sleep (**Fig. S7 on page 5 of this rebuttal letter**). Our results are consistent with the findings that REM sleep was not prolonged after 3-h or 6-h SD in rats. It is reported that REM sleep rebound occurred only when REM sleep has severe deficits [48]. However, although a mild 2 h REM sleep deficiency is induced by chemogenetic inactivation of RMTg GABAergic neurons, there is a significant promotion of REM sleep within the following 4 h from 13:00-17:00.

Therefore, chemogenetic inactivation of RMTg GABAergic neurons induced significant enhancement of REM sleep compared with SD groups. We think the effects of using chemogenetic techniques cannot be compared directly with that using optogenetics.

(2) It is hard to understand the in vivo recording results listed in Fig 4A. Are those spikes recorded from 4 channels of an optrode, or 4 different RMTg GABAergic neurons? In addition, scale bar for this panel was incorrect. For example, 500 ms should be 500 μ m.

Reply:

Those spikes in Fig. 4a were recorded from 4 channels of a tetrode probe. It is the same unit recorded from 4 channels heated and twisted together in one tetrode probe. Sorry, the scale bar was incorrect. We checked and corrected the scale bars in Fig. 4a.

Reviewer #3 (Remarks to the Author):

Zhao et al. found that the rostromedial tegmental nucleus (RMTg) GABAergic (RMTg-GABA) neurons play an essential role in the regulation of rapid eye movement (REM) sleep. Chemogenetic activation of RMTg-GABA neurons decreased amount of REM sleep by decreasing the number of transitions from NREM (NREM) sleep to REM sleep.

Optogenetic activation of RMTg-GABA neurons during REM sleep induced transition to wakefulness (Wake) and then NREM sleep. Optogenetic inhibition of RMTg-GABA neurons during NREM sleep increased transition to REM sleep, and that during REM sleep increased the latency of REM sleep transition. In vivo electrophysiological recording of RMTg-GABA neurons revealed that those neurons are active during both Wake and REM sleep, while the activity was gradually increased during the first 30 s of REM sleep. Additionally, optogenetic activation of nerve terminal of RMTg-GABA in the laterodorsal tegmental (LDT) nucleus induced transition to Wake and then NREM sleep, while that in the lateral hypothalamus (LH) induced direct transition to NREM sleep. In vitro ontogenetic recording revealed that both cholinergic and GABAergic neurons in the LDT receive inhibitory input from RMTg-GABA neurons, and glutamatergic neurons in the LDT receive local inhibition from GABAergic neurons. Finally, optogenetic activation of RMTg-GABA neurons in the LDT where Vglut2 is knocked down during REM sleep induced direct transition into NREM sleep. The authors well examined the novel role of RMTg-GABA neurons which induce transition from REM sleep to Wake or NREM sleep. Particularly, the result that the activity of RMTg-GABA neurons gradually increased during REM sleep indicating reciprocal modulation of REM sleep was exciting. They should add some more information mentioned as follows:

Major comments:

1) In Figure 1D and 1E, the authors showed only REM sleep data. Wake and NREM sleep also should be involved in as Figure to confirm the effect of activation of RMTg-GABA neurons.

Reply:

Thank you for the suggestion. We added the time course of NREM sleep and wakefulness, as well as the amount of NREM sleep and wakefulness in the mice with chemogenetic activation of RMTg GABAergic neurons in **Fig. 1e, f**. When RMTg GABAergic neurons were specifically activated by CNO injection at a dose of 1 mg/kg, the mice showed a significant increase in NREM sleep and a decrease in wakefulness and REM sleep compared to the saline group, which is consistent with the effects when the neurons in the RMTg were nonspecifically activated in rats [10].

This information has been added on lines 113-117, page 6.

Fig. 1e, f After administration of saline or CNO, the hourly average amount of each stage [REM sleep (9:00–17:00): $F_{1,16} = 9.158$, $p = 0.0080$; NREM sleep (9:00–14:00): $F_{1,16} = 14.85$, $p = 0.0014$; wake (9:00–14:00): $F_{1,16} = 7.517$, $p = 0.0145$] (e) and total amount of each stage [REM sleep (9:00–17:00): $T_8 = 3.662$, $p = 0.0064$; NREM sleep (9:00–14:00): $T_8 = 3.435$, $p = 0.0089$; wake (9:00–14:00): $T_8 = 2.674$, $p = 0.0283$] (f)

2) In Figure 2, 3, 4, 5 and 7, the authors performed optogenetic manipulation or electrophysiological recordings and sleep recordings. However, the timing of experiments such as light or dark phase is not described in the methods. The probability of each state can be differ depending on the circadian rhythm. Therefore, they should be careful about the timing and describe it in detail in the methods.

Reply:

Thank you for the suggestion. The optogenetic experiments and *in vivo* electrophysiology experiments were mainly performed during the light period. For REM and NREM sleep stimulations, the percentage of trials stimulated during the dark phase were less than 5% and 10%, respectively. For wake stimulation, the percentage of trials stimulated during the dark phase was from 5% to 32%. For *in vivo* electrophysiology experiments, the recordings performed during the dark period were 26%.

The information of “The photostimulation experiments were mainly performed during the light period” was added in the method section on lines 939-940, page 56.

3) In Figure 2, 3, 5 and 7, the authors performed optogenetic manipulation. They described the timing of photostimulation in lines 932–936, but it is unclear whether the stimuli induced in vigilance state dependent manner or randomly, and what is the interval between the stimuli. Those conditions may affect the sleep architecture.

Reply:

Sorry for our not clear description. The timing of photostimulation was random, either during REM sleep, NREM sleep, and quiet wakefulness or active movement. We found that optogenetic excitation of RMTg GABAergic neurons promoted NREM sleep regardless of quiet or active wakefulness. Since REM sleep occurs every 10 to 20 min in mice [6], the interval between the light stimuli was set for at least 10 min.

The detailed information on optogenetic experiments was added in the method section on lines 939-949, pages 56-57.

4) In Figure S4 experiment, isn't there a possibility that the increase of REM sleep amount within 4 hours (13:00–17:00) is just a rebound of REM sleep decreased during 2-h post-CNO injection period (9:00–11:00)? The results of the following optogenetic inhibition seem to be reliable, but those of the chemogenetic inhibition are hard to agree with the authors' current discussion in lines 600–605. The authors' idea would be described.

Reply:

Thank you for the question. This point is similar to the questions from reviewer 1# and 2#. **Please refer to the replies in detail on page 4, page 26, and Fig. S7 on page 5.**

In all, we conducted an additional experiment to confirm that 2-h sleep deprivation (SD) during the light phase cannot induce any REM sleep rebound. Therefore, that the increase in REM sleep with chemogenetic inhibition of RMTg GABAergic neurons is also reliable, which is consistent with that using optogenetic inhibition.

We added the information in the results on lines 256-267, page 16; and a paragraph in discussion on lines 673-684, page 43.

5) In Figure 5, the proportion of the first transition from REM sleep to NREM and Wake would be involved to clearly show RMTg-GABA → LTD projection mainly induces transition to Wake, while RMTg-GABA → LH projection mainly does that to directly NREM.

Reply:

Thank you for the suggestion. This point is similar to the question from the reviewer 1#. **Please refer to Fig. 7e-g on page 19.** Fig. 7e-g showed the probability of the first transitions from REM sleep to wakefulness (e, f) and REM to NREM sleep (g).

For the RMTg^{GABA}-LH projection, the proportion of the first REM-wake transitions decreased from 96% in the mCherry group to 14% in the Chr2 group (Fig. 7f) and the proportion of the first REM-NREM sleep transitions increased accordingly (Fig. 7g). For the RMTg^{GABA}-LDT projection, the proportion of the first REM-wake

transitions was 88% in the mCherry and 84% in ChR2 group (Fig. 7f), which was similar to that in mice when RMTg GABAergic neurons are stimulated (Fig. 7e). After ablation of glutamatergic LDT neurons, the proportion of the first REM-wake transitions decreased from 98% in the shCtrl group to 2% in the shVglut2 group and proportion of the first REM-NREM sleep transitions increased accordingly (Fig. 7f, g).

In summary, the RMTg^{GABA}-LDT projection mainly induces REM-wake transitions, while the RMTg^{GABA}-LH projection mainly induces direct transitions from REM to NREM sleep. The legend of Fig. 7e-g has been rewritten to clearly show the change of first transitions between different brain states.

Minor comments:

1) Figure S1A “Merge”: The meaning of small white triangles should be described. If the triangles show coexistence of Foxp1 and GABA, there are 2 unnecessary ones in the bottom right.

Reply:

Thank you for the suggestion. The merged cells pointed as white arrows in Fig. S1a showed the coexistence of Foxp1 and GABA in the RMTg (lines 95-96, page 5). Two white triangles in the bottom right were deleted.

2) Figure S1A and S1B: The length of scale bar should be described.

Reply:

Scale bar, 50 μ m in Fig. S1a-c. It was added on the legend.

3) Figure S1C “Transforming” and “Rebuilding”: The method of “transforming” and “rebuilding” should be described in the methods.

Reply:

After Foxp1 was immunostained with green fluorescence, the pictures were captured by a confocal microscope. The images were processed in Image J. The RGB images were converted into 8-bit pictures, where the boundary of the RMTg was defined from the adjacent region above a certain threshold in black & white images. Then automatic wand tool was applied and representative cell bodies were colored and matched to the mouse brain atlas accordingly. The above methods were supplied in the legend text for Fig. S1c.

(4) Figure 1C and S4C: The baseline of the membrane potential and the firing rate is quite different. It seems that command current was injected in either or both experiments. The methods should be described in more detail. Additionally, statistical analysis such as firing rate before and after CNO application would be involved to clearly show the effectivity of chemogenetic manipulation.

Reply:

We observed two types of RMTg GABAergic neurons, some populations have spontaneous spikes, while others have no spontaneous spikes. Among neurons with spontaneous firing, the neurons generally fire at about 5 Hz. During the patch-clamp experiment, we first record as a baseline and then add 5 μ M CNO to excite the hM3Dq-expressing neurons or inhibit hM4Di-expressing neurons in the RMTg.

The data has been statistically analyzed and two scatterplots about firing rates and membrane potentials in hM3Dq and hM4Di, respectively, before and after CNO application were added in Fig. 1c, d and Fig. S4c (now changed Fig. S6c).

Fig. 1c A representative trace recorded from an hM3Dq-expressing RMTg GABAergic neuron during application of clozapine-N-oxide (CNO) in a brain slice showing an increase in the action potential firing. **d** The average firing rates ($T_6 = 7.913$, $p = 0.0002$) and membrane potential ($T_6 = 5.943$, $p = 0.0010$) of RMTg hM3Dq-expressing neurons ($N = 7$ cells from 3 mice) were significantly increased by CNO application. The results from each cell are shown on the scatter plot.

Fig. S6c A typical trace showed that application of clozapine-N-oxide (CNO) inhibited spontaneous firing and decreased the membrane potential of an hM4Di-expressing RMTg GABAergic neuron. The average firing rates ($T_5 = 7.370$, $p = 0.0007$) and membrane potential ($T_5 = 3.347$, $p = 0.0204$) of RMTg hM3Dq-expressing neurons ($N = 6$ cells in 3 mice) were significantly decreased by CNO application. The results from each cell are shown on the scatter plot.

5) Figures showing “Sleep–wake state changes and probability of brain state transitions after photostimulation”, such as Figure 2E: Whether these Figures show data from all animals or a representative animal should be described.

Reply:

Thank you for the question. In figures such as Figure 2E, the data are from all animals in each group. This information has been added to the related figure legends.

6) Figures showing “cumulative probability”, such as Figure 2G: Calculation of the cumulative probability should be described in the methods since those Figures are confusing. For example, Figure 2E “ChR2” showed immediate termination of REM sleep by photostimulation. Therefore, I assumed that “cumulative probability” would immediately increase in the first tens of seconds in Figure 2G “REM termination” (like Weber et al. (2018) Nat commn. Figure 1f “REM->Wake” <https://doi.org/10.1038/s41467-017-02765-w>). However, the actual Figure showed gradual increase of “cumulative probability” through 120 sec.

Reply:

Thank you for your suggestion. We recalculated the cumulative probability according to Weber et al. 2018 [4]. For the cumulative probability analysis of each brain state, firstly we calculated cumulative trial numbers that firstly entered or terminated REM sleep, NREM sleep, or wakefulness at a specific time point from the onset of photostimulation to 120 s in the ChR2 experiment or 60 s in the Arch experiment.

As shown in Fig. 2g, the cumulative initiation probability analysis of each brain state, such as the cumulative probability of wake initiation at a certain time point after the start of laser stimulation was calculated as follows: trials that first entered the awake state plus the trials entered the awake state before this time point, and then divide these total trials by all trials from all mice in either control group or experimental group. Cumulative termination probability analysis of each brain state, such as the cumulative probability of REM sleep termination in Fig. 2g at a certain time point from the onset of laser stimulation was calculated as follows: trials that transit REM sleep to other two brain states plus the trials that transit REM sleep to other two brain states before this time point, and then divide these total trials by all trials from all mice in either control group or experimental group.

We have recalculated all related data and replaced the corresponding figures. Additionally, the calculation of the cumulative probability has been added in the method on lines 956-961, page 57.

7) Lines 191–192: “ChR2 group” and “the control group” are opposite.

Reply:

Sorry for the mistake, which has been corrected.

8) Figure 4A “spike waveforms”: If the waveforms are from individual units, Figures of each waveform should be clearly separated.

Reply:

Spike waveforms in Fig. 4a are not from 4 individual units. Those spikes are from 4 channels of an optrode.

9) Figure 4: Figures of *in vivo* electrophysiological recording sites would be involved.

Reply:

Thank for the suggestion. Fig. 4b showed the recording sites of 12 identified units from 3 mice in our *in vivo* electrophysiological experiment. Each indigo dot indicates one unit.

Fig. 4b Recording sites of 12 identified units across the rostrocaudal extent of the RMTg in VGAT-Cre mice. Each indigo dot indicates one unit.

10) Figure 5H “561 nm”: Before 561 nm stimulus period, is this really REM sleep? The value of delta band of EEG power spectrum seems to be higher compared to that of “473 nm” data below.

Reply:

Thank for the question. As shown in Fig. 5h, the EEG (top) is not very representative. Thus, it has been replaced.

11) Figure 6N: The latency of the response seems too fast in some data which are less than 1 ms compared to other data such as Figure 6D and 6I. Are these responses surely inhibited by picrotoxin? Wasn't there a possibility that ChR2 itself ectopically expressed in the cholinergic neurons?

Reply:

Thank the reviewer for pointing out the error. Due to no efficient virus encoding glutamate reporter gene, we microinjected the virus of AAV-DIO-ChR2-mCherry and AAV-ChAT-eGFP into the LDT of VGAT-Cre mice to explore whether there is a functional connection that LDT GABAergic neurons project to glutamatergic neurons. Thus, in the LDT, GABAergic neurons and their nerve endings express ChR2 (mCherry, red), and cholinergic neurons express eGFP, as a result, non-fluorescent cells are considered glutamatergic neurons. It's reported that stimulation of the presynaptic inputs induced response of postsynaptic neurons greater than 1 ms and less than 5 ms [49,50]. Thus, the cells with a latency of less than 1 ms were mistaken for glutamatergic neurons instead of LDT GABAergic cells expressing a small amount of ChR2.

After being carefully checked, the cells with a latency of less than 1 ms were removed and we performed this experiment again. More viruses were microinjected in the LDT and longer time for the virus expression in mice. When we conducted the

patch-clamp recording, we used higher fluorescence intensity and carefully identified the recorded neurons that did not express mCherry or eGFP. The data has been reanalyzed and the corresponding figure 6k-o (now changed Fig. 6p-t) has been corrected.

Fig. 6p–t Within the LDT, GABAergic terminals formed inhibitory connections with glutamatergic neurons.

12) Figure 7B: Did these slices express ChR2 or not? If ChR2 are expressed, the annotation “Glut” would be incorrect.

Reply:

The coronal sections in Fig. 7b containing the LDT expressed ChR2 only on RMTg GABAergic nerve fibers indicated with green fluorescence, rather than cell bodies. Under this situation, the sections were stained against glutamate antibodies, again indicated with green fluorescence, so the annotation Glut means the cell bodies, not nerve fibers.

This information has been added on lines 595-599, page 39.

13) Line 659: “knockout” should be “knock down”.

Reply:

Sorry for the mistake, which has been deleted.

14) After manipulation of RMTg-GABA neurons to change the sleep architecture, were there any rebounds of each vigilance state?

Reply:

Thank you for the suggestion. The following response figure 4 showed that after the end of 120-s photostimulation, there is a rebound of REM sleep and wakefulness after REM sleep inhibition (a). Similarly, for blue light stimulation during NREM sleep, there is a rebound of REM sleep and wakefulness with a tendency after NREM sleep promotion after the end of photostimulation (Fig. b). However, for blue light stimulation during wakefulness, NREM sleep is still maintained after NREM sleep promotion by exciting RMTg GABAergic neurons (Fig. c).

It has been proposed that REM sleep propensity accumulates in NREM sleep and discharge during REM sleep [37]. Disruption of discharge of REM-sleep propensity by REM sleep inhibition and accumulation of REM-sleep propensity in NREM sleep by optogenetic excitation of RMTg GABAergic neurons may result in REM sleep

rebounds. These results also indicate the activity of RMTg GABAergic neurons is necessary for terminating REM sleep physiologically. These results are in line with findings by Weber, et al (2018), who found that optogenetic manipulation of periaqueductal GABAergic neurons during REM sleep inhibited REM sleep and resulted in REM sleep rebound [4].

While, after laser stimulation with blue light during wakefulness in mice, NREM sleep was generated and the promotion of the NREM sleep was maintained even if the laser stimulation is stopped. These results indicated that the NREM sleep/wake homeostasis mechanism is not the main factor in regulating sleep rebound on a timescale of minutes in rodents [51,52].

Response Figure 4 A rebound of each vigilance state after the end of optogenetic activation of RMTg GABAergic neurons. During 300 s after the end of 120-s laser stimulation of RMTg GABAergic neurons, the probability of REM sleep (left) and wake (right) after REM sleep inhibition in Fig. 2E (a) and after NREM sleep promotion in Fig. 2H (b) and probability of wake (left) and NREM sleep (right) after wake inhibition in Fig. S3a (c). Black, control group. The colorful, experimental group (green, wakefulness; magenta, REM sleep; blue, NREM sleep). Yellow and red, values with significant differences (Two-way ANOVA followed by a post hoc Sidak test). Shading, SEM.

References:

- 1 Wess J, Nakajima K, Jain S. Novel designer receptors to probe GPCR signaling and physiology. *Trends Pharmacol Sci.* 2013;34(7):385-92.
- 2 Eban-Rothschild A, Rothschild G, Giardino WJ, Jones JR, de Lecea L. VTA dopaminergic neurons regulate ethologically relevant sleep-wake behaviors. *Nature neuroscience.* 2016;19(10):1356-66.

- 3 Li YD, Luo YJ, Xu W, Ge J, Cherasse Y, Wang YQ, et al. Ventral pallidal GABAergic neurons control wakefulness associated with motivation through the ventral tegmental pathway. *Molecular psychiatry*. 2021;26(7):2912-28.
- 4 Weber F, Hoang Do JP, Chung S, Beier KT, Bikov M, Saffari Doost M, et al. Regulation of REM and Non-REM Sleep by Periaqueductal GABAergic Neurons. *Nature communications*. 2018;9(1):354.
- 5 Yu X, Li W, Ma Y, Tossell K, Harris JJ, Harding EC, et al. GABA and glutamate neurons in the VTA regulate sleep and wakefulness. *Nature neuroscience*. 2019;22(1):106-19.
- 6 Weber F, Dan Y. Circuit-based interrogation of sleep control. *Nature*. 2016;538(7623):51-59.
- 7 Luo YJ, Li YD, Wang L, Yang SR, Yuan XS, Wang J, et al. Nucleus accumbens controls wakefulness by a subpopulation of neurons expressing dopamine D1 receptors. *Nature communications*. 2018;9(1:1576.):1576.
- 8 Burdakov D, Karnani MM. Ultra-sparse Connectivity within the Lateral Hypothalamus. *Curr Biol*. 2020;30(20):4063-70 e2.
- 9 Dib R, Gervais NJ, Mongrain V. A review of the current state of knowledge on sex differences in sleep and circadian phenotypes in rodents. *Neurobiol Sleep Circadian Rhythms*. 2021;11:100068.
- 10 Yang SR, Hu ZZ, Luo YJ, Zhao YN, Sun HX, Yin D, et al. The rostromedial tegmental nucleus is essential for non-rapid eye movement sleep. *Plos Biol*. 2018;16(4).
- 11 Lu J, Jhou TC, Saper CB. Identification of wake-active dopaminergic neurons in the ventral periaqueductal gray matter. *The Journal of neuroscience : the official journal of the Society for Neuroscience*. 2006;26(1):193-202.
- 12 Hasegawa E, Miyasaka A, Sakurai K, Cherasse Y, Li Y, Sakurai T. Rapid eye movement sleep is initiated by basolateral amygdala dopamine signaling in mice. *Science (New York, NY)*. 2022;375(6584):994-1000.
- 13 Van Dort CJ, Zachs DP, Kenny JD, Zheng S, Goldblum RR, Gelwan NA, et al. Optogenetic activation of cholinergic neurons in the PPT or LDT induces REM sleep. *Proceedings of the National Academy of Sciences of the United States of America*. 2015;112(2):584-9.
- 14 Jogo S, Glasgow SD, Herrera CG, Ekstrand M, Reed SJ, Boyce R, et al. Optogenetic identification of a rapid eye movement sleep modulatory circuit in the hypothalamus. *Nature neuroscience*. 2013;16(11):1637-43.
- 15 Feng H, Wen SY, Qiao QC, Pang YJ, Wang SY, Li HY, et al. Orexin signaling modulates synchronized excitation in the sublateralodorsal tegmental nucleus to stabilize REM sleep. *Nature communications*. 2020;11(1):3661.
- 16 Clement O, Sapin E, Libourel PA, Arthaud S, Brischoux F, Fort P, et al. The lateral hypothalamic area controls paradoxical (REM) sleep by means of descending projections to brainstem GABAergic neurons. *The Journal of neuroscience : the official journal of the Society for Neuroscience*. 2012;32(47):16763-74.
- 17 Feng X, Zhao HY, Shao YJ, Lou HF, Zhu LY, Duan S, et al. Anxiolytic Effect of Increased NREM Sleep after Acute Social Defeat Stress in Mice. *Neurosci Bull*. 2020;36(10):1137-46.
- 18 Yu X, Zhao G, Wang D, Wang S, Li R, Li A, et al. A specific circuit in the midbrain detects stress and induces restorative sleep. *Science (New York, NY)*. 2022;377(6601):63-72.
- 19 Boyce R, Glasgow SD, Williams S, Adamantidis A. Causal evidence for the role of REM

- sleep theta rhythm in contextual memory consolidation. *Science*. 2016;352(6287):812-6.
- 20 Glover EJ, Starr EM, Chao Y, Jhou TC, Chandler LJ. Inhibition of the rostromedial tegmental nucleus reverses alcohol withdrawal-induced anxiety-like behavior. *Neuropsychopharmacology*. 2019;44(11):1896-905.
- 21 Fu R, Zuo W, Shiwalkar N, Mei Q, Fan Q, Chen X, et al. Alcohol withdrawal drives depressive behaviors by activating neurons in the rostromedial tegmental nucleus. *Neuropsychopharmacology*. 2019;44(8):1464-75.
- 22 Arrigoni E, Chee MJS, Fuller PM. To eat or to sleep: That is a lateral hypothalamic question. *Neuropharmacology*. 2019;154:34-49.
- 23 Zhao YN, Zhang Y, Tao SY, Huang ZL, Qu WM, Yang SR. Whole-Brain Monosynaptic Afferents to Rostromedial Tegmental Nucleus Gamma-Aminobutyric Acid-Releasing Neurons in Mice. *Front Neurosci*. 2022;16:914300.
- 24 Liu D, Dan Y. A motor theory of sleep-wake control: arousal-action circuit. *Annu Rev Neurosci*. 2019;42:27-46.
- 25 Dornellas APS, Burnham NW, Luhn KL, Petrucci MV, Thiele TE, Navarro M. Activation of locus coeruleus to rostromedial tegmental nucleus (RMTg) noradrenergic pathway blunts binge-like ethanol drinking and induces aversive responses in mice. *Neuropharmacology*. 2021;199:108797.
- 26 Lahti L, Haugas M, Tikker L, Airavaara M, Voutilainen MH, Anttila J, et al. Differentiation and molecular heterogeneity of inhibitory and excitatory neurons associated with midbrain dopaminergic nuclei. *Development (Cambridge, England)*. 2016;143(3):516-29.
- 27 Smith RJ, Vento PJ, Chao YS, Good CH, Jhou TC. Gene expression and neurochemical characterization of the rostromedial tegmental nucleus (RMTg) in rats and mice. *Brain structure & function*. 2019;224(1):219-38.
- 28 Quina LA, Tempest L, Ng L, Harris JA, Ferguson S, Jhou TC, et al. Efferent pathways of the mouse lateral habenula. *The Journal of comparative neurology*. 2015;523(1):32-60.
- 29 Flint RR, Chang T, Lydic R, Baghdoyan HA. GABA(A) receptors in the pontine reticular formation of C57BL/6J mouse modulate neurochemical, electrographic, and behavioral phenotypes of wakefulness. *The Journal of neuroscience : the official journal of the Society for Neuroscience*. 2010;30(37):12301-9.
- 30 Xi MC, Morales FR, Chase MH. Evidence that wakefulness and REM sleep are controlled by a GABAergic pontine mechanism. *Journal of neurophysiology*. 1999;82(4):2015-9.
- 31 Boissard R, Gervasoni D, Schmidt MH, Barbagli B, Fort P, Luppi PH. The rat ponto-medullary network responsible for paradoxical sleep onset and maintenance: a combined microinjection and functional neuroanatomical study. *Eur J Neurosci*. 2002;16(10):1959-73.
- 32 Sanford LD, Tang X, Xiao J, Ross RJ, Morrison AR. GABAergic regulation of REM sleep in reticularis pontis oralis and caudalis in rats. *Journal of neurophysiology*. 2003;90(2):938-45.
- 33 Xi M, Fung SJ, Yamuy J, Chase MH. Interactions between hypocretinergic and GABAergic systems in the control of activity of neurons in the cat pontine reticular formation. *Neuroscience*. 2015;298:190-9.
- 34 Sun Y, Cao J, Xu C, Liu X, Wang Z, Zhao H. Rostromedial tegmental nucleus-substantia nigra pars compacta circuit mediates aversive and despair behavior in mice. *Exp Neurol*.

- 2020;333:113433.
- 35 Taylor NE, Long H, Pei J, Kukutla P, Phero A, Hadaegh F, et al. The rostromedial tegmental nucleus: a key modulator of pain and opioid analgesia. *Pain*. 2019;160(11):2524-34.
- 36 Polter AM, Barcomb K, Tsuda AC, Kauer JA. Synaptic function and plasticity in identified inhibitory inputs onto VTA dopamine neurons. *Eur J Neurosci*. 2018;47(10):1208-18.
- 37 Benington JH, Heller HC. REM-sleep timing is controlled homeostatically by accumulation of REM-sleep propensity in non-REM sleep. *Am J Physiol*. 1994;266(6 Pt 2):R1992-2000.
- 38 Lu J, Sherman D, Devor M, Saper CB. A putative flip-flop switch for control of REM sleep. *Nature*. 2006;441(7093):589-94.
- 39 Saper CB, Fuller PM, Pedersen NP, Lu J, Scammell TE. Sleep state switching. *Neuron*. 2010;68(6):1023-42.
- 40 Park SH, Weber F. Neural and Homeostatic Regulation of REM Sleep. *Front Psychol*. 2020;11:1662.
- 41 Lazarus M, Chen JF, Huang ZL, Urade Y, Fredholm BB. Adenosine and Sleep. *Handb Exp Pharmacol*. 2019;253:359-81.
- 42 Onorato I, Neuenschwander S, Hoy J, Lima B, Rocha KS, Broggin AC, et al. A Distinct Class of Bursting Neurons with Strong Gamma Synchronization and Stimulus Selectivity in Monkey V1. *Neuron*. 2020;105(1):180-97 e5.
- 43 Schneider M, Broggin AC, Dann B, Tzanou A, Uran C, Sheshadri S, et al. A mechanism for inter-areal coherence through communication based on connectivity and oscillatory power. *Neuron*. 2021;109(24):4050-+.
- 44 Yang Y, Cui Y, Sang K, Dong Y, Ni Z, Ma S, et al. Ketamine blocks bursting in the lateral habenula to rapidly relieve depression. *Nature*. 2018;554(7692):317-22.
- 45 Ferrari LL, Park D, Zhu L, Palmer MR, Broadhurst RY, Arrigoni E. Regulation of Lateral Hypothalamic Orexin Activity by Local GABAergic Neurons. *The Journal of neuroscience : the official journal of the Society for Neuroscience*. 2018;38(6):1588-99.
- 46 Bonnavion P, Jackson AC, Carter ME, de Lecea L. Antagonistic interplay between hypocretin and leptin in the lateral hypothalamus regulates stress responses. *Nature communications*. 2015;6.
- 47 Bonnavion P, Mickelsen LE, Fujita A, de Lecea L, Jackson AC. Hubs and spokes of the lateral hypothalamus: cell types, circuits and behaviour. *J Physiol*. 2016;594(22):6443-62.
- 48 Tobler I, Borbely AA. The Effect of 3-H and 6-H Sleep-Deprivation on Sleep and Eeg Spectra of the Rat. *Behav Brain Res*. 1990;36(1-2):73-78.
- 49 Chuhma N. Functional Connectome Analysis of the Striatum with Optogenetics. *Adv Exp Med Biol*. 2021;1293:417-28.
- 50 Mingote S, Chuhma N, Rayport S. Optogenetic Mapping of Synaptic Connections in Mouse Brain Slices to Definethe Functional Connectome of Identified Neuronal Populations. *Bio Protoc*. 2017;7(1):e2090.
- 51 Tononi G, Cirelli C. Sleep function and synaptic homeostasis. *Sleep Med Rev*. 2006;10(1):49-62.
- 52 Porkka-Heiskanen T. Sleep homeostasis. *Curr Opin Neurobiol*. 2013;23(5):799-805.

REVIEWER COMMENTS

Reviewer #1 (Remarks to the Author):

Thank you for your thoughtful responses to my and the other reviewers comments. The manuscript is much clearer now.

Reviewer #2 (Remarks to the Author):

The paper by Zhao et al. investigates the role of GABAergic neurons in the rostromedial tegmental nucleus (RMTg) in the regulation of REM sleep. The authors conducted many experiments to address this issue, utilizing state of the art genetic circuit dissection techniques and challenging experimental approaches. In the revised manuscript, most of my major concerns have been appropriately addressed. The following points are some of my new concerns, although they are minor. I hope the answers to these questions can be helpful for the publication of this paper.

(1) The authors should quantitatively evaluate the co-localization rate of GABA and Foxp1 in the RMTg (Fig S1). In addition, does the RMTg include other type of neurons? Whether the other RMTg neurons also co-localize with Foxp1? I think these evaluations are fundamental because the authors manipulated the GABAergic neurons of RMTg in most of their experiments.

(2) In Figure 1f, why did the authors analyze the REM sleep data in 9:00-17:00 (8hrs), whereas the NREM and wake data in 9:00-14:00 (5hrs). Please specify it.

(3) In Figure 4a (right panel), the latency between blue light onset and spike activity gradually changes, as illustrated by the time interval between the blue markers and spike dots. Please check it again.

(4) Finally, I hope the language of this paper can be thoroughly checked for grammatical correctness, and the overall readability improved.

Reviewer #3 (Remarks to the Author):

This is a revised manuscript by Zhao et al. The authors have appropriately responded to the previous comments noted by this reviewer. Of particular interest was the response to comment #14: rebound-like phenomena are observed for REM suppression and NREM promotion, but not for Wake suppression.

This reviewer has still some minor comments.

In the response to Major comment #2, the author stated in the Methods that "The photostimulation experiments were mainly performed during the light period.". However, this is not enough to describe what time in day the experiments were performed. It should be stated that what time in the light period the experiments (n=?) and what time in the dark period (n=?).

In the response to Minor comment #4, the authors still do not respond to my concern. I understood that there are two types of RMTg-GABAergic neurons. If so, why the authors could record from lower firing-rate neurons in hM3Dq experiment (Fig. 1c) whereas could do from higher firing-rate neurons in hM4Di experiment (Fig. S6c)? If they intentionally selected neurons for each experiment, they should clarify the criteria of the selection in Methods part.

Point-by-point responses to referees

REVIEWER COMMENTS

Reviewer #1 (Remarks to the Author):

Thank you for your thoughtful responses to my and the other reviewers comments. The manuscript is much clearer now.

Reviewer #2 (Remarks to the Author):

The paper by Zhao et al. investigates the role of GABAergic neurons in the rostromedial tegmental nucleus (RMTg) in the regulation of REM sleep. The authors conducted many experiments to address this issue, utilizing state of the art genetic circuit dissection techniques and challenging experimental approaches. In the revised manuscript, most of my major concerns have been appropriately addressed. The following points are some of my new concerns, although they are minor. I hope the answers to these questions can be helpful for the publication of this paper.

(1) The authors should quantitatively evaluate the co-localization rate of GABA and Foxp1 in the RMTg (Fig S1). In addition, does the RMTg include other type of neurons? Whether the other RMTg neurons also co-localize with Foxp1? I think these evaluations are fundamental because the authors manipulated the GABAergic neurons of RMTg in most of their experiments.

Reply:

Thank you for the suggestion. We quantified the co-localization rate of GABA and Foxp1 in the RMTg for Fig. S1 and found that 96% of Foxp1-expressing neurons (green) were colocalized with GABA (red), and 93% of GABA-positive neurons co-expressed Foxp1. We created a statistical figure as Fig.S1a right.

It has been reported that the RMTg is primarily composed of GABAergic neurons, which accounted for about 70-92% of the total neurons ¹. There is until now no convincing evidence that the RMTg contains other major neural populations aside from GABAergic neurons ^{2, 3}. Moreover, we examined in situ hybridization on the Allen brain (<https://mouse.brain-map.org/experiment/siv?id=73818754&imageId=73844028&initImage=ish&coordSystem=pixel&x=5161&y=3497&z=3>) and found no vesicular glutamate transporter expressed in the RMTg. The results of distinct afferents of the RMTg driving

dissociable responses ⁴ and diverse afferent inputs to RMTg GABAergic neurons ⁵ indicated that the RMTg may include distinct subregions that mediated different functions through special output pathways rather than comprise other types of neurons.

Although RMTg neurons are mainly GABAergic, such neurons are also densely distributed around the RMTg. Fortunately, some studies using RNA sequencing showed that *Foxp1* is specifically expressed at high levels in the mouse RMTg GABAergic neurons, but not in its surrounding area such as VTA GABAergic neurons ^{6,7}. Therefore, we used *Foxp1* as a specific molecular marker to define the location the RMTg and VGAT-Cre mice were used for experiments because of the high co-expression rates of *Foxp1* and GABA.

The information “96% of *Foxp1*-expressing neurons was colocalized with GABA, and 93% of GABA-positive neurons co-expressed *Foxp1* (Fig. S1a, right). As a result, *Foxp1* was used as a specific molecular marker to define the location the RMTg and GABAergic neurons in the RMTg were manipulated using vesicular GABA transporter (VGAT)-Cre mice in this study” was added on lines 97-100, page 5.

Fig. S1 RMTg location defined by *Foxp1* immunostaining on the mouse atlas. a Left panel, immunostaining of a typical brain section stained against *Foxp1* (green) and GABA (red), and co-localized *Foxp1* and GABA staining (yellow, the cells pointed to with white arrows). Top, the red circle depicts the RMTg location. Bottom, higher magnification image of the white box in the top pictures. Right panel, quantification of neurons that co-expressed *Foxp1* and GABA. n = 4 mice.

(2) In Figure 1f, why did the authors analyze the REM sleep data in 9:00-17:00 (8hrs), whereas the NREM and wake data in 9:00-14:00 (5hrs). Please specify it.

Reply:

Thank you for the question. In the present manuscript, we found strong and long-lasting suppression of REM sleep by activation of RMTg-GABA neurons. From the time course of REM sleep (Figure 1e), after clozapine-N-oxide (CNO) injection at 9:00, the hourly amount of REM sleep between 9:00 to 17:00 was less than that of the saline group during the corresponding period. However, at 17:00, REM sleep reached the same level as the saline group. Therefore, we analyzed the sum REM sleep amount during the 8-h post-CNO injection period (Figure 1f). Now we reanalyzed NREM sleep and awake data during the same period (9:00-17:00) and replaced these two sub-figures in Figure 1f. The revised statistics about these two sub-figures have been corrected on lines 145-151, page 9 of this revised manuscript. We found activation of RMTg GABAergic neurons also significantly promoted NREM sleep, which agrees with our previous findings⁸.

Fig. 1e, f During the 8-h period (9:00–17:00) after administration of saline or CNO at 9:00, the hourly average amount of each stage (REM sleep: $F_{1,16} = 9.158$, $p = 0.0080$; NREM sleep: $F_{1,16} = 6.395$, $p = 0.0223$; wake: $F_{1,16} = 2.254$, $p = 0.1527$) (e) and total amount of each stage (REM sleep: $T_8 = 3.662$, $p = 0.0064$; NREM sleep: $T_8 = 2.410$, $p = 0.0425$; wake: $T_8 = 1.662$, $p = 0.1351$) (f).

(3) In Figure 4a (right panel), the latency between blue light onset and spike activity gradually changes, as illustrated by the time interval between the blue markers and spike dots. Please check it again.

Reply:

Thank you for the suggestion. We carefully checked the data in Figure 4a (right panel)

and found that the latency between blue light onset and spike activity did not change. Sorry for the mistake. This figure has been corrected.

Fig. 4 Firing rates of identified RMTg GABAergic neurons across brain states.

a An example unit. Comparison between laser-evoked (blue) and spontaneous (black) spike waveforms from this unit (left). Spike raster showing multiple trials of laser stimulation at 30 Hz. Blue ticks, laser pulses (right).

(4) Finally, I hope the language of this paper can be thoroughly checked for grammatical correctness, and the overall readability improved.

Reply:

Thank you for the suggestion. We asked Editage (www.editage.cn) to thoroughly edit English to correct the grammar and improve readability.

Reviewer #3 (Remarks to the Author):

This is a revised manuscript by Zhao et al. The authors have appropriately responded to the previous comments noted by this reviewer. Of particular interest was the response to comment #14: rebound-like phenomena are observed for REM suppression and NREM promotion, but not for Wake suppression. This reviewer has still some minor comments.

(5) In the response to Major comment #2, the author stated in the Methods that “The photostimulation experiments were mainly performed during the light period.”. However, this is not enough to describe what time in day the experiments were performed. It should be stated that what time in the light period the experiments (n=?) and what time in the dark period (n=?).

Reply:

Thank you for the suggestion. In the optogenetic experiments on sleep-wake behaviors, a laser was randomly delivered to different brain states of NREM and REM sleep and wakefulness. All trials in which light stimulation was initiated when mice naturally entered REM sleep for at least 16 s or NREM sleep/wakefulness for at least

60 s were used for analysis ⁹. All 1705 trials used for analysis were from 37 mice, all of which were given photostimulation during the light period (7:00–19:00). Among these 1705 trials, 125 trials accounting for 7% were from the dark period (19:00–7:00) in 23 of such 37 mice.

The information of “The optogenetic experiments on sleep–wake behaviors were performed during both the light (7:00–19:00) and dark period (19:00–7:00)” was added in the method section on lines 844-845, page 50.

(6) In the response to Minor comment #4, the authors still do not respond to my concern. I understood that there are two types of RMTg-GABAergic neurons. If so, why the authors could record from lower firing-rate neurons in hM3Dq experiment (Fig. 1c) whereas could do from higher firing-rate neurons in hM4Di experiment (Fig. S6c)? If they intentionally selected neurons for each experiment, they should clarify the criteria of the selection in Methods part.

Reply:

Thank you for the question. In the *in vitro* electrophysiological experiments to validate the effectiveness of chemogenetic manipulation of RMTg GABAergic neurons, we did not specifically select neurons for the application of CNO. The patch-clamp experiment is described below:

To verify the chemogenetic manipulation of RMTg GABAergic neurons, we randomly recorded neurons expressing the mCherry reporter gene. In the current clamp mode, with a holding voltage of -60 mV, we first recorded the baseline level and then added 5 μ M CNO to excite the hM3Dq-expressing neurons or inhibit the hM4Di-expressing neurons in the RMTg. The firing frequency and cell membrane potential of the recorded RMTg GABAergic neurons before and after CNO treatment were analyzed. This information has been added on lines 948-954, page 55.

It may be that the number of recorded neurons in the previously revised manuscript is limited ($N = 7$ cells in the M3 experiment from 3 mice; $N = 6$ cells in the M4 experiment from 3 mice), so the firing rate of the RMTg recorded neurons was different at the baseline level between the M3 and M4 experiments. Now we did additional experiments to record more neurons in an attempt to reduce the sampling variability ($N = 13$ cells in the M3 experiment from 5 mice; $N = 15$ cells in the M4 experiment from 5 mice). The data have been combined and we found that there was no significant difference of the baseline firing for both M3 and M4 verification ($T_{26} = 1.707$, $p = 0.10$ by unpaired t test). Now two scatterplots about firing rates and membrane potentials in hM3Dq and hM4Di, respectively, before and after CNO application were revised in Fig. 1d, and Fig. S6c (now changed Fig. S7c).

According to our results, RMTg neurons do have differences in whether they have

spontaneous firing or not and the different frequency of spontaneous firing, however, the patch-clamp recording data indicated that the chemogenetic manipulation of RMTg GABAergic neurons works.

Fig. 1d The average firing rates ($T_{12} = 8.446$, $p < 0.0001$) and membrane potential ($T_{12} = 9.217$, $p < 0.0001$) of RMTg hM3Dq-expressing neurons ($N = 13$ cells from 5 mice) were significantly increased by CNO application. The results from each cell are shown on the scatter plot.

Fig. S7c The average firing rates ($T_{14} = 5.385$, $p < 0.0001$) and membrane potential ($T_{14} = 7.997$, $p < 0.0001$) of RMTg hM4Di-expressing neurons ($N = 15$ cells from 5 mice) were significantly decreased by CNO application. The results from each cell are shown on the scatter plot.

References:

1. Jhou TC, Fields HL, Baxter MG, Saper CB, Holland PC. The Rostromedial Tegmental Nucleus (RMTg), a GABAergic Afferent to Midbrain Dopamine Neurons, Encodes Aversive Stimuli and Inhibits Motor Responses. *Neuron* **61**, 786–800 (2009).
2. Zhao YN, *et al.* The rostromedial tegmental nucleus: anatomical studies and roles in sleep and substance addictions in rats and mice. *Nat Sci Sleep* **12**,

1215–1223 (2020).

3. Jhou TC. The rostromedial tegmental (RMTg) “brake” on dopamine and behavior: A decade of progress but also much unfinished work. *Neuropharmacology* **198**, 108763 (2021).
4. Li H, *et al.* Three Rostromedial Tegmental Afferents Drive Triply Dissociable Aspects of Punishment Learning and Aversive Valence Encoding. *Neuron* **104**, 987–999 e984 (2019).
5. Zhao YN, Zhang Y, Tao SY, Huang ZL, Qu WM, Yang SR. Whole-Brain Monosynaptic Afferents to Rostromedial Tegmental Nucleus Gamma-Aminobutyric Acid-Releasing Neurons in Mice. *Front Neurosci* **16**, 914300 (2022).
6. Lahti L, *et al.* Differentiation and molecular heterogeneity of inhibitory and excitatory neurons associated with midbrain dopaminergic nuclei. *Development (Cambridge, England)* **143**, 516–529 (2016).
7. Smith RJ, Vento PJ, Chao YS, Good CH, Jhou TC. Gene expression and neurochemical characterization of the rostromedial tegmental nucleus (RMTg) in rats and mice. *Brain structure & function* **224**, 219–238 (2019).
8. Yang SR, *et al.* The rostromedial tegmental nucleus is essential for non-rapid eye movement sleep. *Plos Biol* **16**, (2018).
9. Luo YJ, *et al.* Nucleus accumbens controls wakefulness by a subpopulation of neurons expressing dopamine D1 receptors. *Nature communications* **9**, 1576 (2018).

REVIEWER COMMENTS

Reviewer #2 (Remarks to the Author):

The authors addressed all my comments adequately. I do not have any additional suggestions.

Reviewer #3 (Remarks to the Author):

The authors thoroughly answered this reviewers' concerns. The manuscript can be accepted to be published.